# Structural basis for mTORC1 regulation by the CASTOR1–GATOR2 complex

Rachel M. Jansen[1,2], Clément Maghe[1,2], Karla Tapia[1], Selina Wu[1], Serim Yang [1,2], Xuefeng Ren [1,2], Roberto Zoncu [1,2]✉ & James H. Hurley [1,2,3]✉

Mechanistic target of rapamycin complex 1 (mTORC1) is a nutrient-responsive master regulator of metabolism. Amino acids control the recruitment and activation of mTORC1 at the lysosome through the nucleotide loading state of the heterodimeric Rag GTPases. Under low nutrients, including arginine, the GTPase-activating protein complex GATOR1 promotes GTP hydrolysis on RagA/B, inactivating mTORC1. GATOR1 is regulated by the cage-like GATOR2 complex and cytosolic amino acid sensors. To understand how the arginine sensor CASTOR1 binds to GATOR2 to disinhibit GATOR1 under low cytosolic arginine, we determined the cryo-electron microscopy structure of human GATOR2 bound to CASTOR1 in the absence of arginine. Two MIOS WD40 domain β-propellers of the GATOR2 cage engage with both subunits of a single CASTOR1 homodimer. Each propeller binds to a negatively charged MIOS-binding interface on CASTOR1 that is distal to the arginine pocket. The structure shows how arginine-triggered loop ordering in CASTOR1 blocks the MIOS-binding interface, switches off its binding to GATOR2 and, thus, communicates to downstream mTORC1 activation.

mTORC1 is a master integrator of cell-extrinsic signaling and cell-intrinsic nutrient sensing and a master regulator of the cellular balance between anabolism and catabolism[1–4]. As such, dysregulation of mTORC1 activity contributes to numerous cancers and metabolic disorders, making mTOR inhibitors a promising therapeutic strategy[5]. The key step in the activation of mTORC1 is its nutrient-regulated recruitment to the lysosomal membrane by the active Rag GTPase–Ragulator complex[6,7]. The Rag–Ragulator complex is composed of RagA or RagB GTPase, heterodimerized to RagC or RagD and tethered to the membrane by the pentameric Ragulator/LAMTOR complex, whose LAMTOR1 subunit is lipidated[6,8]. In response to nutrients, including amino acids, glucose and cholesterol, the Rag proteins convert between two stable nucleotide states, inactive (RagA$^{GDP}$ or RagB$^{GDP}$:RagC$^{GTP}$ or RagD$^{GTP}$) and active (RagA$^{GTP}$ or RagB$^{GTP}$:RagC$^{GDP}$ or RagD$^{GDP}$)[9–12]. The active Rag dimer is responsible for recruiting mTORC1 to lysosomes[13–16]. When cytosolic amino acid levels are low, the Rag–Ragulator complex is inactivated by the GTPase-activating protein (GAP) GATOR1, which

promotes GTP hydrolysis by RagA or RagB[13]. The activity of GATOR1 is in turn regulated by the protein complexes GATOR2 and KICSTOR[13,17]. The entire system is targeted to the lysosome principally by the Rag–Ragulator complex[18]. GATOR1, GATOR2 and KICSTOR are not known to directly sense amino acids. Instead, a series of dedicated amino acid sensors that include CASTOR1, sestrin2 (SESN2) and SAMTOR relay information about amino acids into the pathway through the intermediation of the GATOR1–GATOR2–KICSTOR complexes[10,19,20]. Understanding how such information is relayed at the structural level is a preeminent question in the regulation of cell metabolism.

GATOR2, a negative regulator of GATOR1, consists of five subunits, WDR59, WDR24, SEH1L, SEC13 and MIOS[13], that come together to form a higher-order cage-like structure that shares components and architectural elements with the COP-II cage and the nuclear pore complex[21]. In their apo states that occur under low amino acids, the arginine sensor CASTOR1 and the leucine sensor SESN2 directly bind to GATOR2, which is thought to prevent the latter from inhibiting the GAP activity of

[1]Department of Molecular and Cell Biology, University of California, Berkeley, Berkeley, CA, USA. [2]California Institute for Quantitative Biosciences, University of California, Berkeley, Berkeley, CA, USA. [3]Helen Wills Neuroscience Institute, University of California, Berkeley, Berkeley, CA, USA. ✉e-mail: rzoncu@berkeley.edu; jimhurley@berkeley.edu

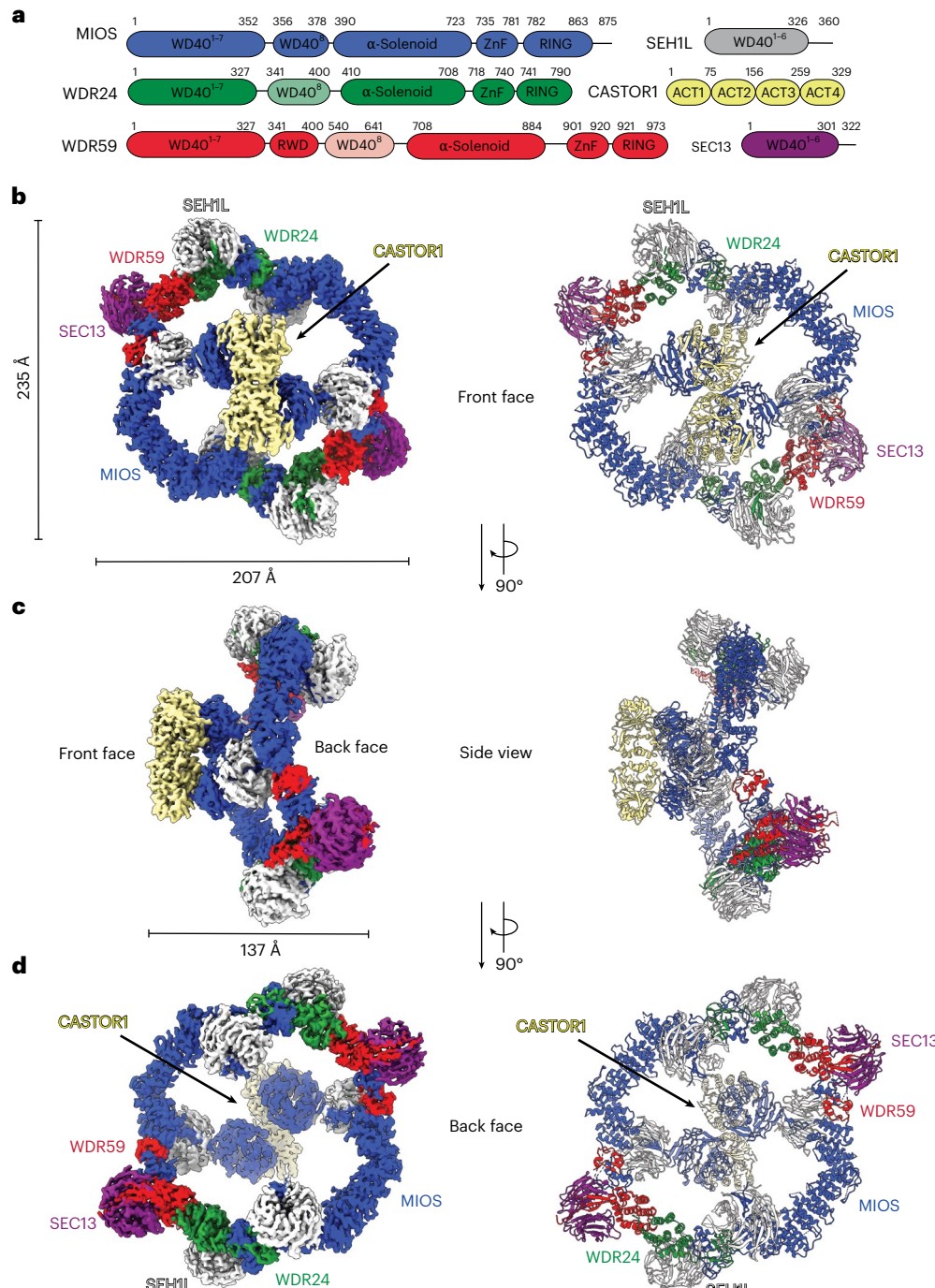

**Fig. 1 | Cryo-EM structure of GATOR2–CASTOR1 complex. a**, Domain organization of subunits within the GATOR2–CASTOR1 structure. **b**–**d**, Composite map and reconstructed model for the GATOR2–CASTOR1 complex viewing from the front face (**b**), side view (**c**) and back face (**d**). Focused maps for different portions of the complex were combined to generate a composite map containing the highest-resolution information for each subunit.

GATOR1 (refs. 10,19,21). The CASTOR1 interaction with arginine triggers the dissociation of CASTOR1 from GATOR2, although the structural mechanism for this step is not yet understood[22]. Previous structural studies uncovered the architecture of GATOR2 and individual nutrient sensors[21–26]. Here, we report the structure of GATOR2 in complex with CASTOR1 in the absence of arginine. By comparing this complex to the preexisting structures of CASTOR1 in the presence and absence of arginine, we were able to deduce and validate the mechanism whereby arginine binding triggers the release of CASTOR1 from GATOR2 by modulating the conformation of a MIOS-releasing loop, thereby regulating the accessibility of the MIOS-binding interface of CASTOR1.

## Results

### Cryo-EM structure of the GATOR2–CASTOR1 complex

To isolate a stable GATOR2–CASTOR1 complex, we purified wild-type GATOR2 from HEK293 cells cotransfected with WDR59, WDR24, SEH1L, SEC13 and MIOS. We separately purified a mutant apo-locked CASTOR1(S111A/D304A) (hereafter referred to as CASTOR1[apo])[22] from an *Escherichia coli* expression system (Extended Data Fig. 1). The purified GATOR2 and CASTOR1[apo] were combined and the cryo-electron microscopy (cryo-EM) structure of GATOR2 bound to CASTOR1[apo] was determined to an overall resolution of 3.89 Å (Fig. 1, Table 1 and Extended Data Fig. 2). The resolution of the complex was further

## Table 1 | Cryo-EM data collection, refinement and validation statistics

| | GATOR2–CASTOR1 complex (EMD-70833), (PDB 9OTI) | GATOR2–CASTOR1–SESN2 |
|---|---|---|
| **Data collection and processing** | | |
| Magnification | 165,000 | 36,000 |
| Voltage (kV) | 300 | 200 |
| Electron exposure (e⁻ per Å²) | 50 | 50 |
| Defocus range (µm) | –1.0 to –2.0 | –1.0 to –2.0 |
| Pixel size (Å) | 0.525 (super-resolution) | 0.525 (super-resolution) |
| Symmetry imposed | None | None |
| Initial particle images (no.) | 2,289,288 | 1,344,786 |
| Final particle images (no.) | 140,606 | 31,364 |
| Map resolution (Å) | 3.02–3.72 | 7.77 |
| FSC threshold | 0.143 | 0.143 |
| **Refinement** | | |
| Initial model used (PDB code) | 7UHY, 5I2C | |
| Model resolution (Å) | 3.89 | |
| FSC threshold | 0.143 | |
| Model resolution range (Å) | 3.24–3.89 | |
| Map sharpening $B$ factor (Å²) | –63 | |
| Model composition | | |
| Nonhydrogen atoms | 43,315 | |
| Protein residues | 6,063 | |
| Ligands | Zn, 29 | |
| $B$ factors (Å²) | | |
| Protein | 74 | |
| R.m.s.d. | | |
| Bond lengths (Å) | 0.002 | |
| Bond angles (°) | 0.453 | |
| **Validation** | | |
| MolProbity score | 1.56 | |
| Clashscore | 8 | |
| Poor rotamers (%) | 0.03 | |
| Ramachandran plot | | |
| Favored (%) | 97.4 | |
| Allowed (%) | 2.6 | |
| Disallowed (%) | 0.03 | |

improved by local refinement resulting in a resolution range of 3.02–3.72 Å (Extended Data Figs. 3 and 4). The cryo-EM density was of sufficient quality to generate an atomistic model of the ordered portions of the core cage of GATOR2 and CASTOR1[apo] (Fig. 1 and Extended Data Fig. 5). The resulting dimensions for the GATOR2–CASTOR1[apo] complex (hereafter simply GATOR2–CASTOR1) were 207 Å × 235 Å × 137 Å.

As seen in the absence of CASTOR1, GATOR2 assembles into an octagonal scaffold containing four copies of MIOS, two copies of WDR24, two copies of WDR59, two copies of SEC13 and six copies of SEH1L (ref. 21). The scaffold is stabilized by four zinc-binding C-terminal domain (CTD) junctions, two of which are formed by MIOS–WDR24 and two of which are formed by MIOS–WDR59, as well as four junctions formed by interactions between the α-solenoid domains of MIOS–MIOS and WDR24–WDR59 (ref. 21) (Fig. 2a). The two MIOS subunits straddling the 'front' face of GATOR2 engage the CASTOR1 homodimer (Fig. 1), while the other two back-facing MIOS β-propellers that do not engage CASTOR1 are disordered and are not seen in the final density map and reconstruction (Figs. 1 and 2a). The $C_2$ symmetry of the GATOR2 complex is broken upon CASTOR1 engagement. Analysis of the refined coordinates revealed that the $C_2$ symmetry of the unbound GATOR2 cage is broken in the CASTOR1-bound complex (Extended Data Fig. 6), although the two asymmetric units differ by a root-mean-square deviation (r.m.s.d.) of only 1.2 Å for Cα atoms.

### CASTOR1 triggers structural rearrangements in GATOR2

The MIOS subunits in GATOR2 have an integral role in the organization of the overall complex core. Comparison with the GATOR2[apo] unbound structure shows that engagement of CASTOR1 with the front-face MIOS β-propellers triggers conformational changes throughout the GATOR2 structure (Fig. 2a,b and Supplementary Video 1). Specifically, upon interaction with CASTOR1, the front-face MIOS β-propeller pair breaks the internal interface, pushing 10 Å apart and rotating by 16° relative to the MIOS α-solenoid domains of the inner core of the GATOR2 cage (Fig. 2a,b) In the GATOR2 unbound conformation, the MIOS interface responsible for interaction with CASTOR1 is exposed and not buried in the β-propeller interface. However, the MIOS β-propellers are too close together to engage both CASTOR1 monomers simultaneously and, thus, must reorient in the bound conformation. The back-face MIOS β-propellers, in contrast, are already separated by a 20-Å gap in unbound GATOR2 (ref. 21) (Fig. 2b). This gap is too far apart to engage the CASTOR1 dimer; therefore, they do not interact with CASTOR1 in the bound complex (Fig. 2b).

The MIOS subunits are intimately linked with the WDR59 subunits through the MIOS–WDR59 CTD junctions in the GATOR2 complex (Extended Data Fig. 7a). In the GATOR2–CASTOR1 complex, the MIOS β-propeller reorientation pushes the front-face SEH1L[MIOS] subunits outward ~8 Å and shifts the MIOS α-solenoid and CTD domains. Compared to GATOR2[apo], this movement in the front-face MIOS α-solenoid and CTD domains results in the disordering of residues 757–836 and 890–921 in WDR59 and residues 727–746 and 770–783 in MIOS (Extended Data Fig. 7b). These residues include the zinc-finger (ZnF) motif in MIOS and WDR59 (Extended Data Fig. 7b). In the GATOR2[apo] unbound structure, the ZnF, along with the RING domains in MIOS and WDR59, stabilizes the CTD–CTD junctions. Specifically, the MIOS ZnF interacts with Sec13 and the WDR59 ZnF interacts with SEH1L (ref. 21). The ZnF contacts are no longer present in the CASTOR1-bound state. However, the RING domains remain intact, preserving the integrity of the GATOR2 cage (Extended Data Fig. 7a).

### The CASTOR1–GATOR2 interface

In the GATOR2-bound structure, a single CASTOR1[apo] homodimer binds across one face of the GATOR2 cage, engaging one MIOS β-propeller domain per CASTOR1[apo] monomer (Fig. 3a,b). The organization of the GATOR2-bound CASTOR1[apo] dimer is unaltered as compared to the previous CASTOR1[apo] crystal structure[26]. The CASTOR1[apo] bound to GATOR2 and the isolated CASTOR1[apo] crystal structure only differ by 0.7 Å r.m.s.d. for Cα atoms. Each CASTOR1 monomer of consists of four ACT (aspartate kinase, chorismate mutase and TyrA) domains that interact through the interface composed of the ACT1 and ACT4 domains[22,24,25]. The two CASTOR1 monomers in the GATOR2-bound CASTOR1 dimer remain equivalent and the two MIOS-binding interfaces as defined by cryo-EM (Fig. 3a) are essentially superimposable on one another. Each interface buries 690 Å² of solvent-accessible surface area.

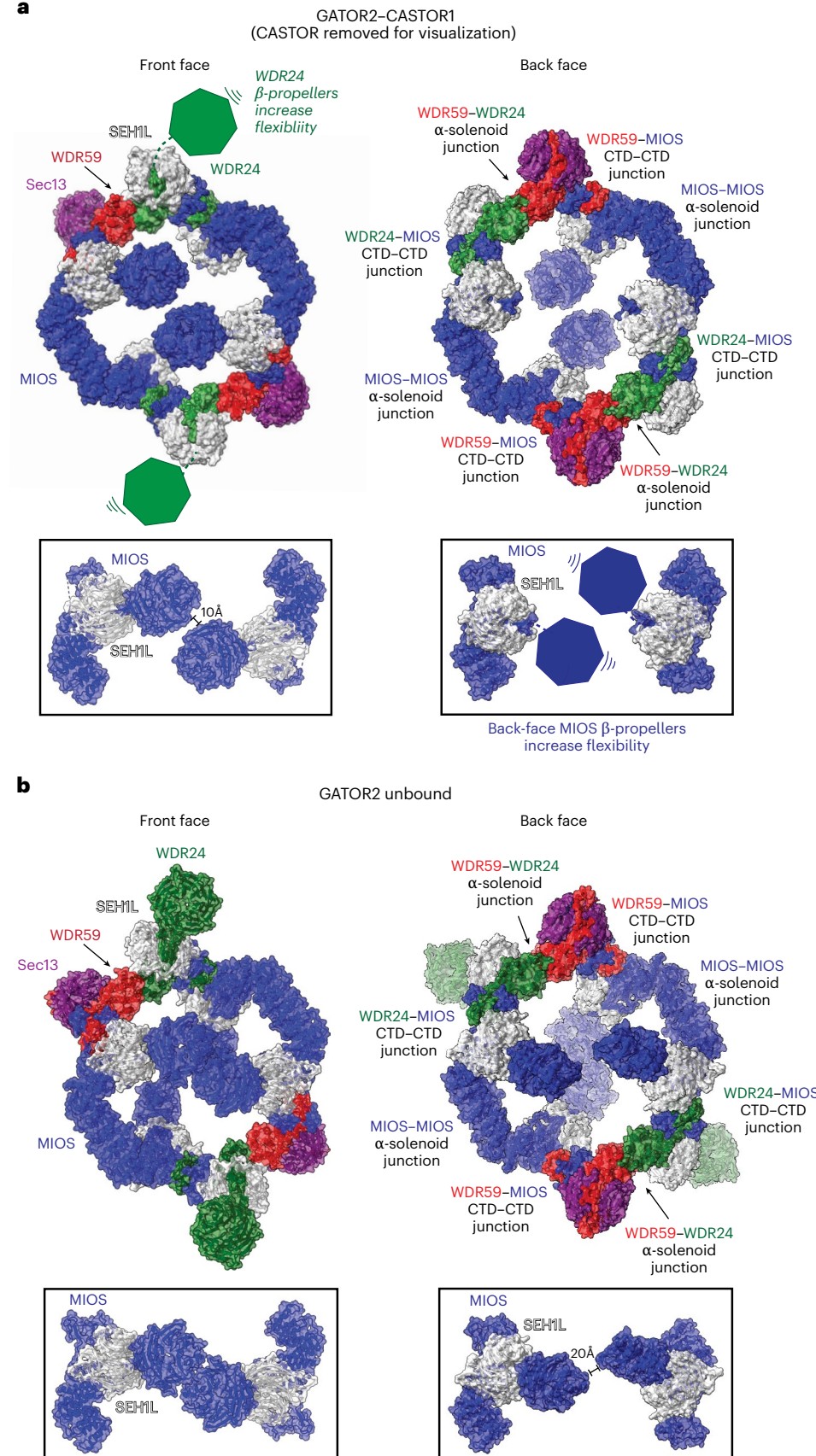

**Fig. 2 | CASTOR1 triggers a structural rearrangement in GATOR2 complex. a,b,** Comparison of the front and back faces of the GATOR2–CASTOR1 (**a**) complex and (**b**) GATOR2$^{apo}$ complex (**b**). CASTOR1 is removed for visualization in the GATOR2–CASTOR1 complex. Changes in the MIOS subunits are highlighted in boxes below complex. Key junctions connecting the inner cage are indicated.

Two MIOS loops are responsible for most of the contacts with CASTOR1[apo] (Fig. 3b,c). Loop 1 (residues 110–114) is located between blades 2 and 3 in the MIOS β-propeller, and loop 2 (residues 134–140) connects two β-sheets in blade 3 (Fig. 3c). The MIOS loops contacts are centered on the basic residues H112, H136 and R137, with K135 at the edge of the contact region (Fig. 3d–f). These four basic residues engage with a complementary electronegative pocket on the surface of CASTOR1[apo] centered on D121 and also containing Y118, Q119 and Y236 (Fig. 3f). To validate the role of the MIOS-binding interface in arginine sensing, mutants were generated within the CASTOR1 pocket (D121A, Y118A, Q119A and W236A) and on the two MIOS loops (H112, K135A, H136A and R137A). The CASTOR1 residues D121, Y118 and Q119 and MIOS residue R137 were previously noted to be important for GATOR2 interaction[22,27]. We transiently expressed wild-type (Flag-tagged) CASTOR1 or CASTOR1 containing single substitutions (Y118A, Q119A, D121A or Y236A) in HEK293T cells. As previously reported, in arginine-deprived cells wild-type CASTOR1–Flag interacted strongly with endogenous GATOR2 (revealed by immunoblotting for MIOS and WDR59), and this interaction was weakened by arginine refeeding[10] (Fig. 3g). The interaction between GATOR2 and CASTOR1 was disrupted in cells expressing mutants within the core of the MIOS-binding interface on CASTOR1—that is, Y118A, Q119A and D121A mutants—as well as cells expressing the mutant Y236A on the periphery of the MIOS-binding interface (Fig. 3g). Next, we coexpressed wild-type Flag-tagged MIOS or l MIOS-binding interface mutants (H112A, K135A, H136A and R137A) with wild-type CASTOR1–HA in HEK239T cells. Substitutions in MIOS loop 1 (H112A) and loop 2 (H136A and R137A) that are central to the MIOS-binding interface disrupted the GATOR2 interaction with CASTOR1 (Fig. 3h). Substituting a MIOS residue on the periphery of the MIOS-binding interface (K135A) had no noticeable effect as compared to wild-type MIOS in arginine-starved conditions. However, the MIOS(R135A) mutant underwent a more complete dissociation from CASTOR1–HA than the wild-type protein upon arginine supplementation, suggesting partial destabilization of the interaction with CASTOR1 by this substitution (Fig. 3h). These data validate the functional relevance of both sides of the CASTOR1–MIOS interface.

To understand the role of the CASTOR1–MIOS interaction on downstream mTORC1 activity, we monitored the phosphorylation of the mTORC1 substrate S6K1 in HEK293T cells depleted of endogenous CASTOR1 through short hairpin RNA (shRNA)-mediated knockdown and reconstituted with transiently expressing CASTOR1 or CASTOR1 containing single substitutions (Y118A, Q119A, D121A or Y236A). shRNA-mediated knockdown of CASTOR1 (Extended Data Fig. 10) rendered mTORC1 partially resistant to arginine deprivation, as shown by enhanced phosphorylation of HA-tagged S6K1 in the arginine-depleted sample (Fig. 3i). Coexpressing wild-type CASTOR1–Flag restored the suppression of HA–S6K1 phosphorylation by arginine depletion. In contrast to wild-type CASTOR1 and consistent with their inability to bind to GATOR2, the CASTOR1 mutants Y118A, Q119A, D121 and Y236A failed to restore the normal pattern of HA–S6K1 phosphorylation by arginine (Fig. 3i).

## Mechanism of arginine-induced CASTOR1 dissociation from GATOR2

The MIOS-binding interface on CASTOR1[apo] is located distal to the arginine-binding pocket (Fig. 4a). To understand how information about arginine levels is communicated between the arginine-binding pocket and MIOS-binding interface, we compared the CASTOR1[apo] structure obtained through the complex of GATOR2–CASTOR to the crystal structure of CASTOR1 bound to arginine[22] (Protein Data Bank (PDB) accession 5I2C) (hereafter referred to as CASTOR1[Arg]) as overlaid on the GATOR2 complex. Examining the electrostatic surface pattern of CASTOR1[apo] and CASTOR1[Arg] revealed that only CASTOR1[apo] has an intact MIOS-binding interface for interaction with MIOS (Fig. 4b,c).

We term the CASTOR1 loop consisting of residues 86–94, which connects β6 and α3 of the ACT2 domain, as the MIOS-releasing loop. In CASTOR1[apo], the MIOS-releasing loop is disordered, which exposes the negatively charged MIOS-binding interface residues Y118, Q119, D121 and Y236 (Fig. 4b). In the CASTOR1[Arg] structure, residues 90–94 in the MIOS-releasing loop are ordered, cover the MIOS-binding interface and sterically block the MIOS–CASTOR1 interaction (Fig. 4c). In essence, the MIOS-releasing loop acts as a lid for the MIOS-binding interface (Fig. 4a). Disordering of the MIOS-releasing loop was previously seen in the isolated CASTOR1[apo] crystal structure; however, the functional and mechanistic relevance of these residues was not explored[26].

To test our structural hypothesis that the MIOS-releasing loop is responsible for dissociating CASTOR1 from MIOS upon arginine binding, we replaced the MIOS-releasing loop with a poly(glycine) segment of equal length, CASTOR1[86–94G], which was designed to be disordered constitutively. The structural hypothesis predicts that the MIOS-binding interface of CASTOR1[86–94G] would remain exposed and functional for MIOS binding even in the presence of arginine. Consistent with the prediction, in HEK293T cells, overexpression of the CASTOR1[86–94G] mutant constitutively bound to MIOS and suppressed mTORC1 phosphorylation of S6K irrespective of arginine levels (Fig. 4d,e).

To explain at the structural level how the negatively charged pocket in CASTOR1[apo] is linked to binding of arginine in the arginine-binding pocket on the other side of CASTOR1, we overlaid the CASTOR1[apo] and CASTOR1[Arg] structures. The global architecture of the proteins remained similar, with rotations observed in the α-helices in the ACT2 and ACT4 domains, the portion of CASTOR1 that consists of the arginine-binding pocket (Fig. 4f–h). The structural comparison shows how adjustments in the ACT2 and ACT4 domains can transmit the arginine signal from the arginine pocket on one side of the CASTOR1 monomer to the MIOS-releasing loop on the other side of the protein.

## GATOR2 interaction with CASTOR1 and SESN2

The leucine sensor SESN2 works in parallel with CASTOR1 to inhibit the activity of GATOR2 and activate GATOR1 (refs. 9,19) when cellular amino acid levels are low. The cryo-EM structure of GATOR2–CASTOR1 revealed the proposed binding sites for SESN2 and GATOR1 on WDR24 and WDR59, respectively, remained free. To visualize how CASTOR1 and SESN2 simultaneously engage with GATOR2, we purified wild-type GATOR2, wild-type GATOR1, CASTOR1[apo] and a mutant apo-locked SESN2(E451Q/R390A/W444E) (hereafter referred to as SESN2[apo])[22,23] from an *E. coli* expression system (Extended Data Fig. 1) to generate a cryo-EM sample of GATOR2, GATOR1, SESN2 and CASTOR1.

A density map was generated for the complex of GATOR2–CASTOR1–SESN2 (Extended Data Fig. 8a–d). GATOR1 was present in the sample and was visualized in the data processing but was not visualized as bound to the GATOR2–CASTOR1–SESN2 complex (Extended Data Fig. 8e). GATOR2–CASTOR1[apo] docked into the final map, suggesting that SESN2 was compatible with the GATOR2–CASTOR1[apo] cage alterations (Extended Data Fig. 9a). Unassigned density was visible in the map at the location of the WDR24 β-propeller and adjacent to it (Extended Data Fig. 9a). We generated an AlphaFold model of SESN2 with a portion of one GATOR2 asymmetric unit containing one copy of WDR24, two copies of SEH1L and one copy of MIOS (Extended Data Fig. 9c). The AlphaFold model was fitted to the cryo-EM map through alignment with the SEH1L subunit connected to WDR24 in GATOR2–CASTOR1 (Extended Data Fig. 9b). In the AlphaFold model, SESN2 makes notable contact with the WDR24 β-propeller, as suggested by previous studies[23,27]. The interface in the AlphaFold model was analyzed and R228 of WDR24 made critical contacts with the negatively charged SESN2 surface (Extended Data Fig. 9d,e). To validate this interaction, a mutant was generated in the WDR24 β-propeller (R228D). We transiently expressed wild-type HA-tagged SESN2 along with wild-type

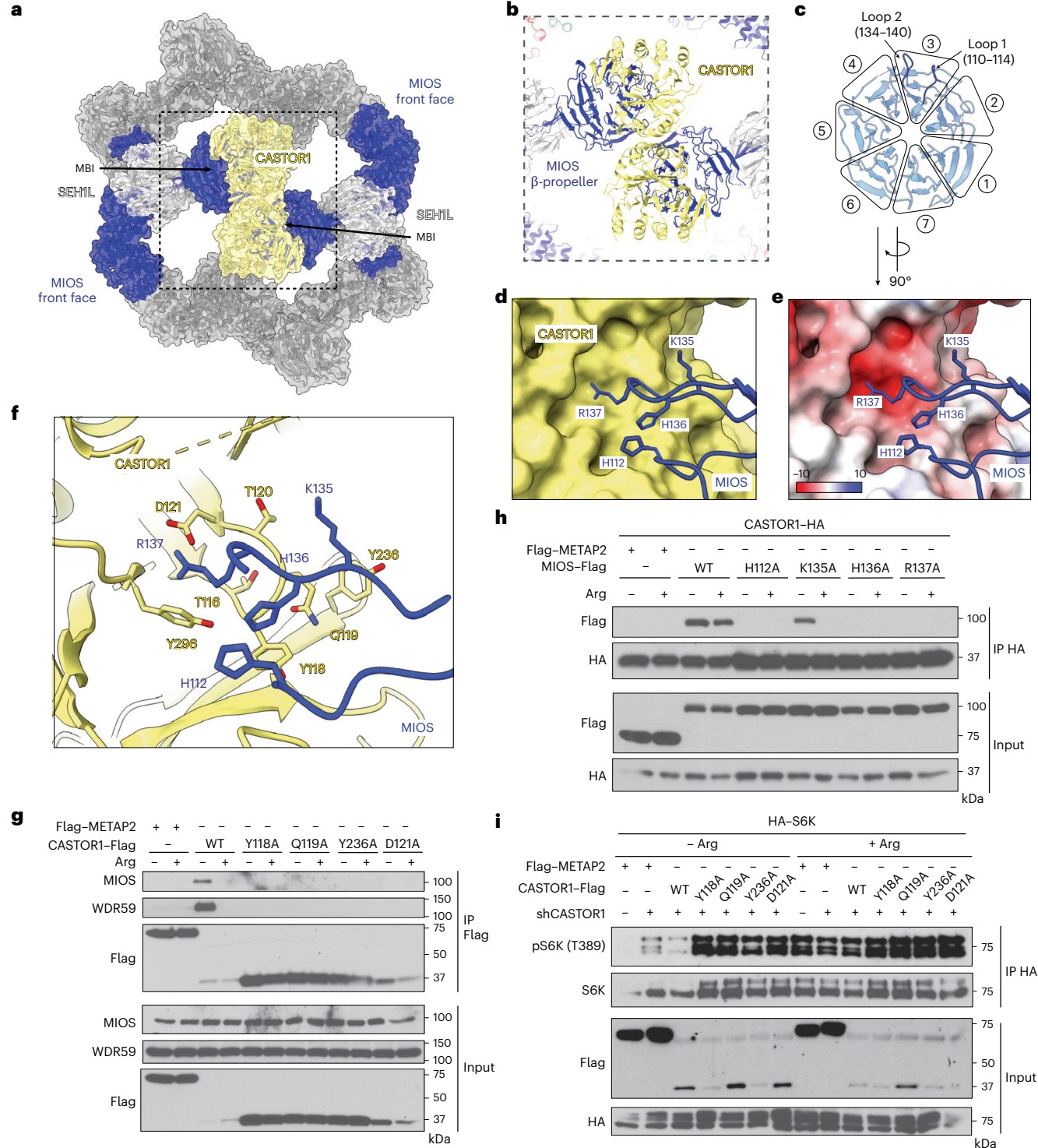

**Fig. 3 | CASTOR1 interacts with MIOS through negatively charged pocket.**
**a**, Overview of GATOR2–CASTOR1 complex. Front-face MIOS subunits (blue) interact with CASTOR1 (yellow). **b**, Close-up view of CASTOR1 interaction with MIOS β-propellers. **c**, Blade diagram for a front-face MIOS β-propeller, highlighting CASTOR1-interacting loops. **d**,**e**, Close-up views of the CASTOR1–MIOS interaction shown with CASTOR1 surface view and MIOS ribbon view (**d**) and CASTOR1 surface colored on the basis of electrostatic potential (**e**). The units of the scale are kcal (mol·e)$^{-1}$ at 298 K. **f**, Ribbon view highlighting specific residues in MIOS loops residues 110–114 and 134–140 (blue) interacting with CASTOR1 residues (yellow). **g**, HEK293T cells transiently expressing the indicated Flag-tagged wild-type (WT) and MIOS-binding interface (MBI)-mutant CASTOR1 constructs or Flag-tagged METAP2 as a control were starved of arginine for 50 min and, where indicated, restimulated for 10 min.

Flag immunoprecipitates (IP) were generated and analyzed by immunoblotting for the indicated proteins. **h**, HEK293T cells transiently expressing CASTOR1–HA and Flag-tagged wild-type MIOS, Flag-tagged MBI-mutant MIOS constructs or Flag-tagged METAP2 as a control. Cells were starved of arginine for 50 min and, where indicated, restimulated for 10 min. Hemagglutinin (HA) immunoprecipitates were generated and analyzed by immunoblotting for the indicated proteins. **i**, CASTOR1-knockdown HEK293T cells transiently expressing the indicated Flag-tagged wild-type and MBI-mutant CASTOR1 constructs or Flag-tagged METAP2 as a control were starved of arginine for 50 min and, where indicated, restimulated for 10 min. Anti-HA immunoprecipitates were prepared and analyzed by immunoblotting for the indicated proteins and phospho-proteins. All cell-based assays were performed three times with similar results.

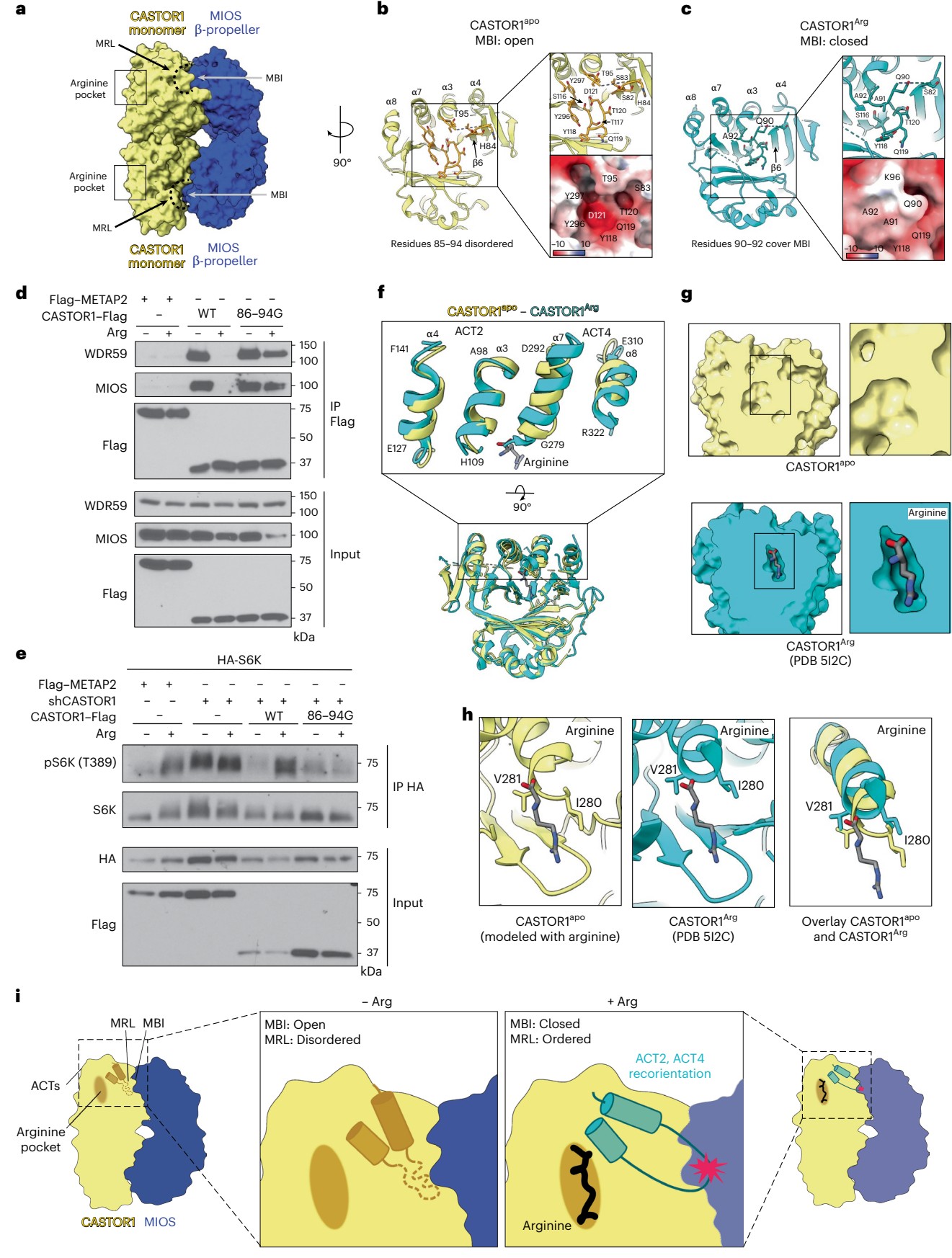

**Fig. 4 | CASTOR1 interaction with arginine triggers closing of GATOR2-interacting pocket. a**, Diagram of CASTOR1 interaction with MIOS β-propellers and location of arginine pocket and MIOS-binding interface. **b**, Electrostatic surface cartoon of CASTOR1[apo] and close-up view of GATOR2-interact pocket. The units of the scale are kcal (mol·e)$^{-1}$ at 298 K. Key residues in CASTOR1 that form the pocket are indicated. **c**, Electrostatic surface cartoon of CASTOR1[Arg] and close-up view of GATOR2-interacting pocket. The units of the scale are kcal (mol·e)$^{-1}$ at 298 K. Key residues in CASTOR1 that block the pocket are indicated. **d**, HEK293T cells transiently expressing the indicated Flag-tagged wild-type and MIOS-releasing loop (MRL)-mutant CASTOR1 constructs or Flag-tagged METAP2 as a control were starved of arginine for 50 min and, where indicated, restimulated for 10 min. Flag immunoprecipitates were generated and analyzed by immunoblotting for the indicated proteins. **e**, CASTOR1-knockdown HEK293T cells transiently expressing the indicated Flag-tagged wild-type and MRL-mutant CASTOR1 constructs or Flag-tagged METAP2 as a control were starved of arginine for 50 min and, where indicated, restimulated for 10 min. Anti-HA immunoprecipitates were prepared and analyzed by immunoblotting for the indicated proteins and phospho-proteins. **f**, Overlay of CASTOR1[apo] (yellow) and CASTOR1[Arg] (cyan). Rotations in ACT2 and ACT4 α-helices are enlarged for visualization. **g**, Surface view of CASTOR1[apo] and CASTOR1[Arg] arginine-binding pocket. CASTOR1[apo] is modeled with arginine in the binding pocket. **h**, Ribbon view of arginine-binding pocket in CASTOR1[apo] and CASTOR1[Arg]. **i**, Overall model for arginine-dependent CASTOR1 interaction with GATOR2. All cell-based assays were performed three times with similar results.

Flag-tagged WDR24 or WDR24 containing single substitution (R228D) in HEK293T cells.

As previously reported, in leucine-deprived cells, wild-type HA–SESN2 interacted strongly with the GATOR2 subunit WDR24 and this interaction was weakened by leucine refeeding[10] (Extended Data Fig. 9f). The interaction between GATOR2 and SESN2 was disrupted in cells expressing WDR24(R228D), suggesting the relevance of the interface observed in the AlphaFold model (Extended Data Fig. 9f). Our cryo-EM and AlphaFold models revealed the location of SESN2 binding that is compatible with the CASTOR1 interaction with GATOR2. Although we only observed one stably bound copy of SESN2 by cryo-EM, it remains possible that an additional copy of SESN2 could interact with the cage, given the second copy of WDR24. Together, these data show that SESN2 and CASTOR1 bind to the same conformational state of GATOR2.

## Discussion

The structure presented here is consistent with a model that links the CASTOR1 interaction with arginine to changes in the GATOR2–CASTOR1 interaction and reveals a mechanism for arginine-induced dissociation of CASTOR1 from GATOR2 leading to mTORC1 activation. Here, we directly visualized the CASTOR1 MIOS-binding interface. Previous structural comparison of isolated apo and arginine-bound CASTOR1 crystal structures noted two missing loop regions in the apo CASTOR1 structure[26]. The functional implications of this change were previously unclear but can now be understood in light of the structure of the GATOR2–CASTOR1 complex. The arginine-binding pocket and the MIOS-binding interface reside on opposite faces of CASTOR1 and are connected by the α-helices of the ACT2 and ACT4 domains. In low-arginine conditions, the GATOR2 pocket is exposed while the arginine-binding pocket is covered. Upon increases in arginine levels, arginine enters the binding pocket and signals through conformational changes in the α-helices to the opposite face of CASTOR1. This leads to ordering of the MIOS-releasing loop, occlusion of the MIOS-binding interface and, thus, the release of CASTOR1 from GATOR2 (Fig. 4i).

We found that one CASTOR1 dimer engages two MIOS WD40 domains on the front face of GATOR2 even though two other MIOS subunits are present on the back face of the cage. The inability of CASTOR1 to bind to the back-face MIOS dimer is explained by the greater separation of these domains. At 20 Å apart in the unbound GATOR2 structure, it may be sterically impossible to draw the back-face MIOS β-propeller pair together to the 10 Å separation needed to bind the CASTOR1 dimer. This prevents the formation of a 2:4 GATOR2 asymmetric unit–CASTOR1 monomer complex. Thus, while the overall cage remains intact, conformational changes extend over the entire cage and break exact $C_2$ symmetry.

The critical remaining question is how the arginine signal is transduced to GATOR1. In yeast, the counterparts of GATOR1 and GATOR2 (the SEA complex) interact directly. The cryo-EM structure of the SEA has been determined[28], yet the precise mechanism of GATOR1 GAP regulation is still unclear, even in yeast. A third protein complex,

KICSTOR, is present in mammals that does not exist in yeast[17]. KICSTOR has been shown to engage both GATOR1 and GATOR2 and regulate their activity[17,29]. The structure of the GATOR2–CASTOR1–SESN2 complex determined here shows that these factors can bind simultaneously, a result consistent with the expectation that, physiologically, low-nutrient states should involve simultaneous depletion of multiple amino acid species. Now that the key question as to how amino acid binding regulates sensor engagement has been answered, at least for CASTOR1 and arginine, the central question going forward is how GATOR1 GAP activity is regulated by the combined action of GATOR2–CASTOR1–SESN2 and KICSTOR. GATOR2 interactions with SESN2, CASTOR1 and GATOR1 are not mutually exclusive and the findings here, thus, set the stage to ultimately answer this question.

How the Rag GTPases interconvert between the active and inactive nucleotide states[9–12] is at the very heart of understanding nutrient regulation of mTORC1. The nucleotide state of RagC and RagD is important primarily for the regulation of noncanonical mTORC1 substrates, of which the MiT-TFE transcription factors are the best characterized[30]. The structural pathway for regulation of the RagC and RagD nucleotide state by the FLCN–FNIP GAP complex has been worked out in large part[31–34]. By contrast, despite its critical importance for both canonical and noncanonical mTORC1 signaling, regulation of the nucleotide state of RagA and RagB remains incompletely understood. Structural analysis of the GATOR1 GAP complex[35,36] and GATOR2 (ref. 21) is enabling strides toward a full structural and mechanistic explanation of this central event. The work presented here adds another important piece to the puzzle, bringing us that much closer to a complete structural view of how the RagA and RagB branch of mTORC1 signaling is regulated.

## Online content

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

## Methods

### Cloning and protein purification

**GATOR2 purification.** Codon-optimized DNA encoding all five subunits of GATOR2 (MIOS, WDR59, WDR24, SEH1L and Sec13) was synthesized by Twist Biosciences and subcloned into the pCAG vector. The construct with MIOS was engineered to include an N-terminal tandem Strep–Flag tag. HEK293-GNTI cells were cotransfected with 1 mg of DNA with an equal amount of all five GATOR2 subunits and 4 mg of PEI per liter of cells at $2 \times 10^6$ cells per ml. Cells were harvested after 48 h and pelleted at 2,000$g$ for 20 min at 4 °C.

Cell pellets were resuspended in 30 ml of lysis buffer (25 mM HEPES pH 7.5, 500 mM NaCl, 2 mM MgCl$_2$, 10% glycerol, 1 mM TCEP, one protease inhibitor tablet (Roche) per 50 ml and 1 mM PMSF) and dounce-homogenized before 1 h of incubation with 1% DDM and CHS (1:10) at 4 °C. The lysate was centrifuged at 37,000$g$ for 35 min at 4 °C. The supernatant was incubated with ~3–4 ml of Strep-Tactin Sepharose resin for 12–15 h with rocking at 4 °C. The resin was washed with 20 ml of high-salt wash buffer A (25 mM HEPES, 500 mM NaCl, 2 mM MgCl$_2$, 1 mM TCEP, 50 mM arginine, 50 mM glutamic acid, 1 mM ATP and 0.03% DDM–CHS, pH 7.4), 20 ml of low-salt wash buffer B (25 mM HEPES, 200 mM NaCl, 2 mM MgCl$_2$, 1 mM TCEP, 50 mM arginine, 50 mM glutamic acid, 1 mM ATP and 0.03% DDM–CHS, pH 7.4), 20 ml of low-salt (no ATP) wash buffer C (25 mM HEPES, 200 mM NaCl, 2 mM MgCl$_2$, 1 mM TCEP, 50 mM arginine, 50 mM glutamic acid and 0.03% DDM–CHS, pH 7.4) and 20 ml of low-salt (no ATP or no DDM–CHS) wash buffer D (25 mM HEPES, 200 mM NaCl, 2 mM MgCl$_2$, 1 mM TCEP, 50 mM arginine and 50 mM glutamic acid, pH 7.4). GATOR2 was eluted from the Strep-Tactin Sepharose resin using 20 ml of elution buffer (25 mM HEPES, 200 mM NaCl, 2 mM MgCl$_2$, 1 mM TCEP, 50 mM arginine, 50 mM glutamic acid and 4 mM desthiobiotin, pH 7.4). Eluted protein was concentrated to 1 ml using a Millipore Amicon Ultra centrifugal filter and subjected to gel filtration using a Superose 6 Increase 10/300 column and buffer containing 25 mM HEPES, 200 mM NaCl, 2 mM MgCl$_2$ and 1 mM TCEP.

**CASTOR1$^{apo}$ purification.** Codon-optimized DNA encoding CASTOR1(S111A/D304A) was synthesized by Twist Biosciences and subcloned into the pET-28a+ vector containing an N-terminal 6×His tag. The vector containing 6×His–CASTOR1$^{apo}$ was transformed into BL21(DE3) cells. Cells were grown at 37 °C until the optical density (OD) reached 0.6. Protein production was induced using 0.2 mM IPTG at 18 °C for 14–16 h. Cells were pelleted by centrifugation at 3,500$g$ for 20 min.

Cell pellets were resuspended in ~50 ml of lysis buffer (30 mM Tris-HCL, 200 mM NaCl, 1 mM TCEP and 1 mM PMSF) and lysed by sonication for 5 min (2 s on and 2 s off). The lysate was centrifuged at 37,000$g$ for 35 min at 4 °C. The supernatant was incubated with ~3 ml of HisPur Ni-NTA resin (Thermo Scientific) for 1–2 h with rocking at 4 °C. The resin was washed with ~150 ml of wash buffer (30 mM Tris-HCL, 200 mM NaCl, 30 mM imidazole and 1 mM TCEP) before elution with ~80 ml of elution buffer (30 mM Tris-HCL, 200 mM NaCl, 200 mM imidazole and 1 mM TCEP). The protein was concentrated using a Millipore Amicon Ultra centrifugation filter to 1 ml. The concentrated protein was subjected to gel filtration using a HiLoad 16/600 Superdex 200 pg column and buffer containing 10 mM HEPES pH 7.5, 100 mM NaCl and 0.5 mM TCEP.

**SESN2$^{apo}$ purification.** Codon-optimized DNA encoding SESN2(E451Q/R390A/W444E) was synthesized by Twist Biosciences and subcloned into the pET-28a+ vector containing an N-terminal 6×His tag. The vector containing 6×His–SESN2$^{apo}$ was transformed into BL21(DE3) cells. Cells were grown at 37 °C until the OD reached 0.7. Protein production was induced using 0.2 mM IPTG at 18 °C for 14–16 h. Cells were pelleted by centrifugation at 3,500$g$ for 20 min.

Cell pellets were resuspended in ~50 ml of lysis buffer (50 mM potassium phosphate pH 8.0, 500 mM NaCl, 30 mM imidazole, 3 mM β-mercaptoethanol (BME) and 1 mM PMSF) and lysed by sonication for 5 min (2 s on and 2 s off). The lysate was centrifuged at 37,000$g$ for 35 min at 4 °C. The supernatant was passed through ~5 ml of HisPur Ni-NTA resin (Thermo Scientific), collected and passed through twice more. The resin was washed with ~150 ml of wash buffer (50 mM potassium phosphate pH 8.0, 500 mM NaCl, 30 mM imidazole, 3 mM BME and 1 mM PMSF) before elution with ~50 ml of elution buffer (50 mM potassium phosphate pH 8.0, 150 mM NaCl, 250 mM imidazole and 3 mM BME). The protein was dialyzed using SnakeSkin dialysis tubing (10 kDa molecular weight cutoff; Thermo Scientific) in 4 l of buffer containing 10 mM potassium phosphate and 100 mM NaCl at 4 °C for 14–16 h. The protein was passed through a 5-ml HiTrap SP HP cation-exchange column (Cytiva) and the flowthrough was collected and saved. The protein was concentrated using a Millipore Amicon Ultra centrifugation filter to 1 ml. The concentrated protein was subjected to gel filtration using a HiLoad 16/600 Superdex 200 pg column and buffer containing 10 mM Tris-HCl pH 8.0, 150 mM NaCl, 0.1 mM EDTA and 0.5 mM TCEP.

**GATOR1 purification.** HEK293-GNTI cells were cotransfected with 1 mg of DNA encoding the GATOR1 subunits glutathione *S*-transferase (GST)-tagged DEPDC5, NPRL2 and NPRL2 at a 1:2:2 ratio and 4 mg of PEI per liter of cells at $2 \times 10^6$ cells per ml. Cells were harvested after 48 h and pelleted at 2,000$g$ for 20 min at 4 °C. Cell pellets were resuspended in 30 ml of lysis buffer (25 mM HEPES pH 7.5, 130 mM NaCl, 2.5 mM MgCl$_2$, 2 mM EGTA, 1% Triton X-100, 0.5 mM TCEP and one protease inhibitor tablet (Roche) per 50 ml) and incubated for 1 h at 4 °C. The lysate was centrifuged at 37,000$g$ for 35 min at 4 °C. The supernatant was incubated with ~3–4 ml of glutathione Sepharose resin for 3 h with rocking at 4 °C. The resin was washed with 15 ml of lysis buffer, 15 ml of high-salt lysis buffer (25 mM HEPES pH 7.5, 500 mM NaCl, 2.5 mM MgCl$_2$, 2 mM EGTA, 1% Triton X-100 and 0.5 mM TCEP), 10 ml of lysis buffer and 15 ml of gel filtration buffer (25 mM HEPES pH 7.5, 130 mM NaCl, 2.5 mM MgCl$_2$ and 0.5 mM TCEP). The column was sealed and an additional 5 ml of gel filtration and tobacco etch virus (TEV) protease was added. The column was incubated with TEV protease overnight for cleavage. The protein was eluted from the column with 15 ml of gel filtration buffer and concentrated to 1 ml using a Millipore Amicon Ultra centrifugal filter. The sample was subjected to gel filtration using a Superose 6 Increase 10/300 column for a final polishing step with buffer containing 25 mM HEPES pH 7.5, 130 mM NaCl, 2 mM MgCl$_2$ and 0.5 mM TCEP.

### Cryo-EM grid preparation and imaging

**GATOR2–CASTOR1$^{apo}$.** Purified GATOR2 was concentrated to 0.45 mg ml$^{-1}$. A threefold molar excess of CASTOR1 was added, incubated for 45 min on ice and immediately frozen on grids. Then, 3 µl of sample was deposited onto freshly glow-discharged (PELCO easiGlow, 30 s in air at 15 mA and 0.4 mbar) holey carbon grids (C-flat: 2/1-3Cu-T). FEI Vitrobot Mark IV was used to blot grids for 3 s with a blot force of 20 (Whatman 597 filter paper) at 4 °C and 100% humidity and subsequently plunged into liquid ethane. The Titan Krios G3i microscope equipped with a Gatan Quantum energy filter (slit width: 20 eV) and a K3 summit camera at a defocus of −1.0 to −2.0 µm was used to record 11,950 videos. Automated image acquisition was performed using SerialEM[37] recording four videos per 2-µm hole with image shift. Image parameters are summarized in Table 1.

**GATOR2–CASTOR1$^{apo}$–SESN2$^{apo}$–GATOR1.** Purified GATOR2 was concentrated to 0.45 mg ml$^{-1}$. A threefold molar excess of CASTOR1, twofold molar excess of SESN2 and threefold molar excess of GATOR1 were added, incubated for 45 min on ice and immediately frozen on grids. Then, 3 µl of sample was deposited onto freshly glow-discharged (PELCO easiGlow, 30 s in air at 15 mA and 0.4 mbar) holey carbon grids (C-flat: 2/1-3Cu-T). FEI Vitrobot Mark IV was used to blot grids for 3 s with a blot force of 20 (Whatman 597 filter paper) at 4 °C and 100% humidity and subsequentially plunged into liquid ethane. The Talos Arctica microscope equipped with a Gatan K3 camera at a defocus of −1.0 to

−2.0 μm was used to record 3,931 videos. Automated image acquisition was performed using SerialEM[37] recording two videos per 2-μm hole with image shift. Image parameters are summarized in Table 1.

## Cryo-EM data processing

The data processing workflow for GATOR2–CASTOR1[apo] is summarized in Extended Data Fig. 1. In short, raw videos were imported into cryoSPARC2 (version 4.3.1)[38]. Patch motion correction was used for motion correction and Patch contrast transfer function (CTF) estimated (multi) was used for CTF determination. The cryoSPARC blob picker with a diameter range of 200–280 Å was used to generate 3,467,659 particles, which were inspected to trim the particle set to 2,289,288 particles. Particles were extracted with a box size of 560 × 560 pixels in cryoSPARC2. A series of two-dimensional (2D) classifications followed by an ab initio reconstruction were used to generate three reference maps. The resulting three-dimensional maps were used in addition to maps generated from prior datasets to resort all 2,289,288 particles after a round of 2D classification to remove obvious 'junk'. The final particle set contained 140,606 particles and a round of homogeneous refinement resulted in a 3.89-Å map at 0.143 Fourier shell correlation (FSC). Masks were generated surrounding various subunits within the complex using UCSF ChimeraX and imported into cryoSPARC2 version 3.3.1, where they were lowpass-filtered and dilated[39] (Extended Data Fig. 2). The masks were used for subsequent local refinement and resulted in improvements of the map between 3.02 and 3.72 Å (Extended Data Figs. 2 and 3) and the generation of a composite map.

The data processing workflow for GATOR2–CASTOR1[apo]–SESN2[apo]–GATOR1 is summarized in Extended Data Fig. 9. In short, raw videos were imported into cryoSPARC2 (version 4.3.1)[37]. Patch motion correction was used for motion correction and Patch CTF estimated (multi) was used for CTF determination. The cryoSPARC blob picker with diameter ranges of 180–230 Å, 210–260 Å and 240–300 Å was used to generate 1,344,786 particles, which were extracted with a box size of 560 × 560 pixels in cryoSPARC2. Volumes from GATOR2–CASTOR1[apo] corresponding to full complex and junk classes were imported and used for subsequent rounds of heterogenous refinement. The final particle set contained 31,364 particles and a round of homogeneous refinement resulted in a 7.77-Å map at 0.143 FSC. The final map revealed density for SESN2 bound to the GATOR2–CASTOR1[apo] cage but not GATOR1. Then, 2D classification was used to visualize the quality of the final particle set. Additionally, the particles picked using 210–260 Å were sorted in 2D for GATOR1 particles. The 2D classes corresponding to GATOR1 were visualized but not bound to the GATOR2 complex.

## Atomic model building and refinement

A composite map for GATOR2–CASTOR1 was generated in UCSF ChimeraX[39] by aligning the local refinement maps to the overall map and combining the best portions of the maps. Focused refinement was performed for all regions in the complex overlapping all subunit. Sections were selected to incorporate all subunits and include sufficient area to generate the highest resolution (Extended Data Figs. 3 and 4). The coordinates for GATOR2 (PDB 7UHY) and arginine-bound CASTOR1 (PDB 5I2C) were rigid-body fitted into the composite map in UCSF ChimeraX[39]. To account for movement of the GATOR2 subunits, the structure was separated into its individual subunits and each subunit was rigid-body fitted independently into the map. The MIOS subunit undergoes the largest conformational change upon CASTOR1 binding. Because of this, the MIOS subunits of GATOR2 were broken down into three smaller portions encompassing residues 43–380, 387–728 and 783–863. Each of these smaller portions was rigid-body fitted into the map. The rigid-body-fitted subunits were combined into another model for further refinement. The model was refined using iterative rounds of PHENIX real-space refinement[40–42]. In between rounds of refinement, the model was manually inspected for fit in the composite map. Residues outside of the map region were manually removed using

Coot. The CASTOR1 substitutions (S111A and D304A) were manually incorporated following the first iteration of refinement using Coot.

## Arginine-binding pocket analysis

Analysis of the CASTOR1 arginine-binding pockets was performed using the CASTp program[43].

## Structure prediction using AlphaFold 3

**GATOR2–CASTOR1–SESN2 prediction.** The structural model of SESN2, WDR24, MIOS and two copies of SEH1L was generated using AlphaFold 3 (ref. 44). The confidence of the predicted models were assessed by predicted local distance difference test. The SESN2–WDR24–MIOS–SEH1L–SEH1L structure was overlaid with each WDR24 subunit of the GATOR2–CASTOR1 cryo-EM structure to generate a GATOR2–CASTOR1–SESN2 full complex prediction.

## Antibodies and chemicals

Antibodies to MIOS (13557S), WDR59 (53385S), Flag (14793S), HA (3724S), S6K1 (2708S) and phospho-T389-S6K1 (9234S) were obtained from Cell Signaling Technology. Antibodies were used at the following dilutions: MIOS, 1:1,000 (Cell Signaling Technology, 13557S, clone D12C6, lot 1); WDR59, 1:1,000 (Cell Signaling Technology, 53385S, clone D4Z7A, lot 1); Flag, 1:1,000 (Cell Signaling Technology, 14793S, clone D6W5B, lot 7); HA, 1:1,000 (Cell Signaling Technology, 3724S, clone C29F4, lot 11); S6K1, 1:1,000 (Cell Signaling Technology, 2708S, clone 49D7, lot 8); phospho-T389-S6K1, 1:1,000 (Cell Signaling Technology, 9234S, clone 108D2, lot 16).

Flag–M2 affinity gel (A2220) and individual powders of amino acids were obtained from Sigma-Aldrich. Pierce anti-HA magnetic beads (88836), Pierce protease inhibitor tablets, EDTA-free (A32965) and hygromycin B (10687010) were obtained from Thermo Fisher Scientific. RPMI1640 medium without glucose and amino acids (R9010-01) was obtained from US Biologicals.

## Mammalian Cell Culture

Adherent HEK293T human embryonic kidney cells were cultured in DMEM supplemented with 10% (v/v) heat-inactivated FBS, penicillin (100 U per ml) and streptomycin (100 μg ml$^{-1}$). Cells were maintained in a humid atmosphere at 37 °C and 5% CO$_2$. Cells were routinely tested for *Mycoplasma* contamination using the MycoAlert *Mycoplasma* detection kit (Lonza, LT07-318).

## Lentivirus production and infection

Lentiviruses were prepared by cotransfecting pLKO.1 constructs along with psPAX2 and pMD2G packaging plasmids into HEK293T cells using the PEI transfection method. Viral supernatant was collected 48 h after transfection and filtered using a 0.45-μm PES syringe filter. The virus was then concentrated using Lenti-X concentrator (Takara Bio, 631232) according to the manufacturer's protocol and stored at −80 °C.

shRNAs directed against CASTOR1 (TRCN0000284010) or Luciferase (TRCN0000072243, used as a nontargeting control) were cloned into the pLKO.1 lentiviral vector (RNAi Consortium, Broad Institute) according to the manufacturer's instructions.

For lentivirus infection, HEK293T cells were seeded along with concentrated virus and 8 μg ml$^{-1}$ polybrene (Millipore, TR-1003-G). After 24 h, the medium was changed to fresh medium supplemented with hygromycin B for selection. Experiments were performed 7 days after infection.

## Transfections, amino acid starvation, cell lysis, immunoprecipitation and western blot

**CASTOR1 interaction with GATOR2.** Transient transfection of cDNA into HEK293T cells was performed using the calcium phosphate transfection method. Briefly, 2 × 10$^6$ HEK293T cells were plated in 10-cm dishes. Then, 24 h later, cells were transfected with the appropriate pRK5-based cDNA in the following amounts: 2,000 ng of METAP2,

3,000 ng of Flag–MIOS, 2,000 ng of CASTOR1–Flag and 2 ng of HA–S6K. The total amount of plasmid DNA was normalized to 5,000 ng with empty pRK5 for each transfection. Then, 6 h later, medium containing the transfection mix was replaced with fresh medium. Experiments were performed 36 h later.

For arginine starvation or restimulation, cells were incubated with arginine-free RPMI for 50 min and, when indicated, restimulated with 1.15 mM arginine for 10 min.

After the indicated treatments, cells were rinsed once with ice-cold PBS and lysed in lysis buffer (10 mM sodium pyrophosphate, 10 mM sodium β-glycerophosphate, 40 mM HEPES, 4 mM EDTA and 1% Triton X-100, pH 7.4, supplemented with one EDTA-free protease inhibitor tablet per 50 ml). After 30 min at 4 °C under gentle agitation, cell lysates were cleared by centrifugation at 17,000$g$ for 10 min at 4 °C. Protein concentrations were normalized across samples using the BCA assay. Equal amounts of proteins were incubated with 30 μl of prewashed anti-HA magnetic beads or Flag–M2 affinity gel for 2 h at 4 °C with end-over-end rotation. The immunoprecipitates were washed three times with lysis buffer before denaturation by the addition of 50 μl of sample buffer and incubation at room temperature for 16 h, 65 °C for 10 min or 95 °C for 5 min. Samples were resolved by 4–20% SDS–PAGE and analyzed by immunoblotting.

**SESN2 interaction with GATOR2.** Transient transfection of cDNA into HEK293T cells was performed using the calcium phosphate transfection method. In brief, $2 \times 10^6$ HEK293T cells were plated in 10-cm dishes. Then, 24 h later, cells were transfected with the appropriate pRK5-based cDNA in the following amounts: 1,000 ng of METAP2, 3,000 ng of Flag–MIOS, 4,000 ng of Flag–WDR24, 500 ng of HA–SESN2, 2,000 ng of CASTOR1–Flag, 2,000 ng of CASTOR1–Flag and 2 ng of HA–S6K. The total amount of plasmid DNA was normalized to 5,000 ng with empty pRK5 for each transfection. Then, 6 h later, medium containing the transfection mix was replaced with fresh medium. Experiments were performed 36 h later. For leucine starvation, cells were incubated with leucine-free RPMI for 50 min. For restimulation, leucine (0.38 mM) was added to the lysates for 2 h during immunoprecipitation.

**cDNA cloning**
Codon-optimized and shRNA-resistant gene fragments (Twist Biosciences) for CASTOR1–Flag and Flag–MIOS were cloned into the pRK5 vector. CASTOR1 and MIOS mutants were generated using the site-directed mutagenesis QuikChange method. In brief, two overlapping primers containing the desired mutation in the center were designed. After PCR amplification, products were DpnI-digested and transformed into chemically competent *E. coli*. Mutations were confirmed by Sanger sequencing (Quintara Biosciences).

**qPCR confirmation shCASTOR1**
RNA was extracted from HEK293T cells using the Aurum total RNA mini kit (Bio-Rad, 732-6820). Equal amounts of RNA were reverse-transcribed using the iScript reverse transcription supermix kit (Bio-Rad, 177-8840). The resulting cDNA was amplified by qPCR using the SsoAdvanced Universal SYBR green supermix (Bio-Rad, 172-5270). Data were analyzed using the $2^{-\Delta\Delta Ct}$ method and normalized by the housekeeping genes *ACTB* and *HPRT1*.

The following primers were used: *ACTB* forward, 5′-GGACTTCGAG CAAGAGATGG-3′; *ACTB* reverse, 5′-AGCACTGTGTTGGCGTACAG-3′; *HPRT1* forward, 5′-TGACACTGGCAAAACAATGCA-3′; *HPRT1* reverse, 5′-GG TCCTTTTCACCAGCAAGCT-3′; *CASTOR1* forward, 5′-GCCACCACC CTCATAGATGT-3′; *CASTOR1* reverse, 5′-AGGAGGTCACTGGGGAACTT-3′.

**Statistics and reproducibility**
Statistical analyses were performed using unpaired two-tailed Student's *t*-tests using GraphPad Prism version 10.2.0 (335) (GraphPad Software). The levels of statistical significance are indicated by asterisks

and the exact *P* values are indicated in each figure legend along with the statistical tests. All cell-based experiments were performed three times independently with similar results unless otherwise specified in the figure legend.

**Reporting summary**
Further information on research design is available in the Nature Portfolio Reporting Summary linked to this article.

## Data availability
The coordinates and density map for the CASTOR1 complex were deposited to the PDB and Electron Microscopy Data Bank (EMDB) with accession codes 9OTI and EMD-70833, respectively. Because the resolution of the of the SESN2 map was only 7.77 Å, we did not deposit it to the EMDB; however, the map and the associated AlphaFold-predicted structure are available from figshare (https://doi.org/10.6084/m9.figshare.29433656)[45]. All other data supporting the findings of this study are available from the corresponding authors on reasonable request. Source data are provided with this paper.

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

## Acknowledgements
We thank Z. Cui and A. Joiner for discussions, R. Hooy for workstation support, D. Fracchiolla for assistance with figure preparation and D. Toso and R. Thakkar for EM support. This work was supported by the National Institutes of Health (R01 CA285366 to J.H.H. and 1R35GM149302 to R.Z.) and a National Science Foundation Graduate Research Fellowship (R.M.J.).

## Author contributions
R.M.J. and J.H.H. conceptualized and designed the research. R.M.J., C.M., K.T., S.W., S.Y. and X.R. carried out the research. R.Z. and J.H.H. supervised the research. R.M.J. and J.H.H. wrote the first draft. All authors edited the manuscript.

## Competing interests
J.H.H. is a cofounder and shareholder of Casma Therapeutics, receives research funding from Hoffmann-La Roche and has consulted for Corsalex. R.Z. is a cofounder and shareholder of Frontier Medicines

and a science advisory board member for Nine Square Therapeutics and receives research funding from Genentech. The other authors declare no competing interests.

## Additional information

**Extended data** is available for this paper at https://doi.org/10.1038/s41594-025-01635-0.

**Correspondence and requests for materials** should be addressed to Roberto Zoncu or James H. Hurley.

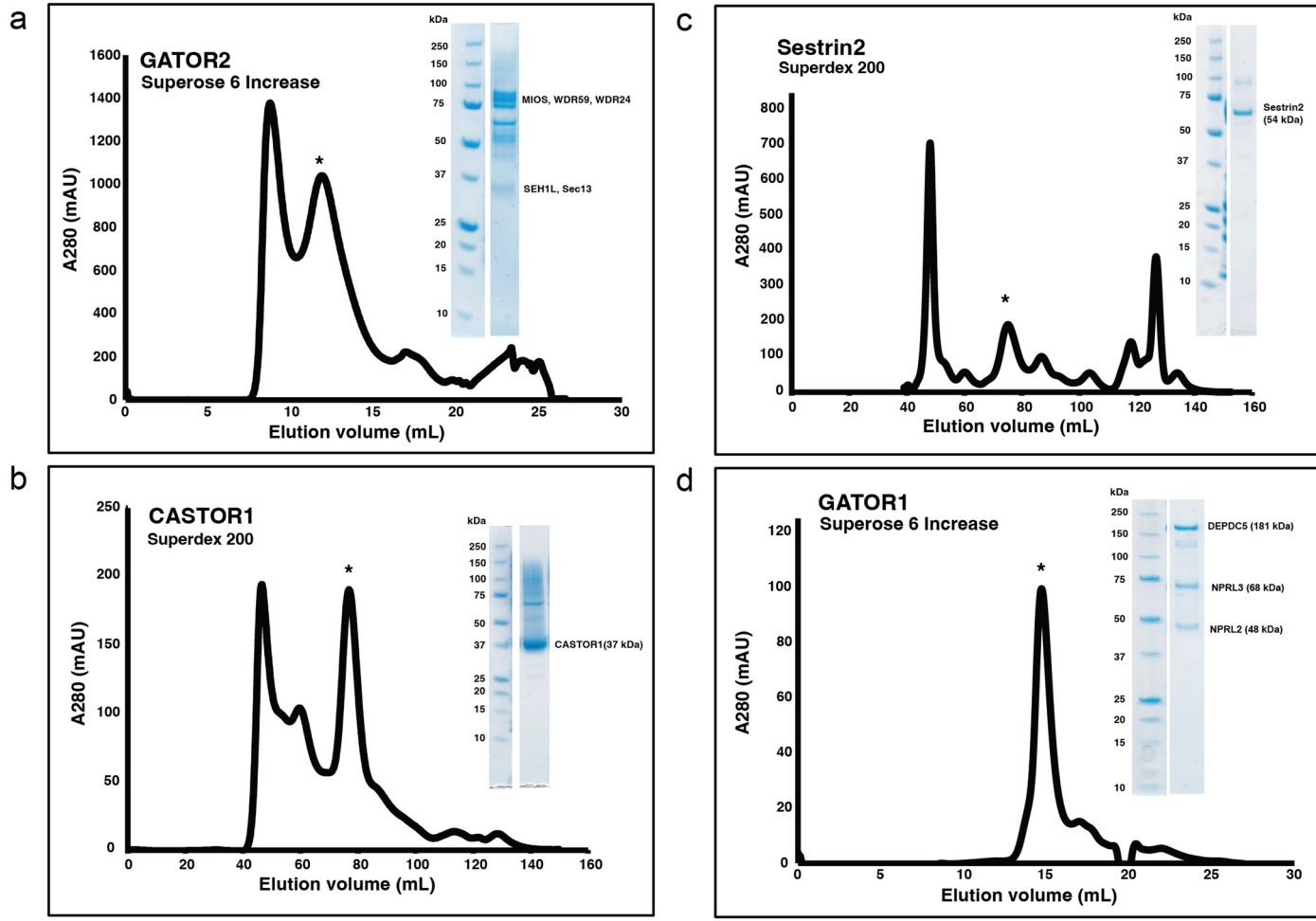

**Extended Data Fig. 1 | Purification for GATOR2 and CASTOR1.** (**a**) Chromatogram and gel for GATOR2 purification. (**b**) Chromatogram and gel for CASTOR1 purification. (**c**) Chromatogram and gel for Sestrin2 purification. (**d**) Chromatogram and gel for GATOR1 purification.

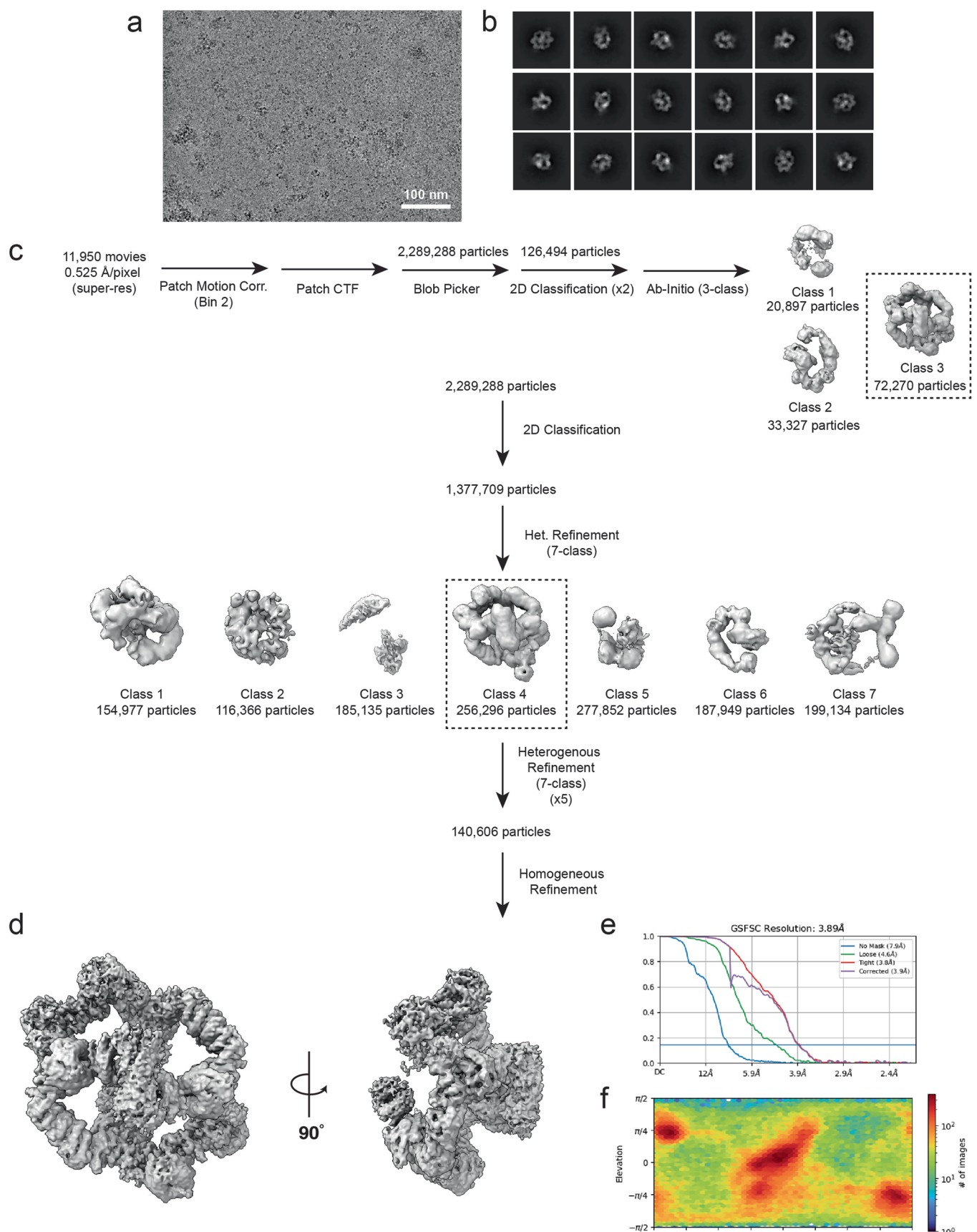

**Extended Data Fig. 2 | Data Processing Pipeline for GATOR2-CASTOR1 complex.** (**a**) Representative micrograph (**b**) Representative 2D classes (**c**) Data processing workflow (**d**) Overall map for GATOR2-CASTOR1 (**e**) FSC graph (**f**) Orientation plot.

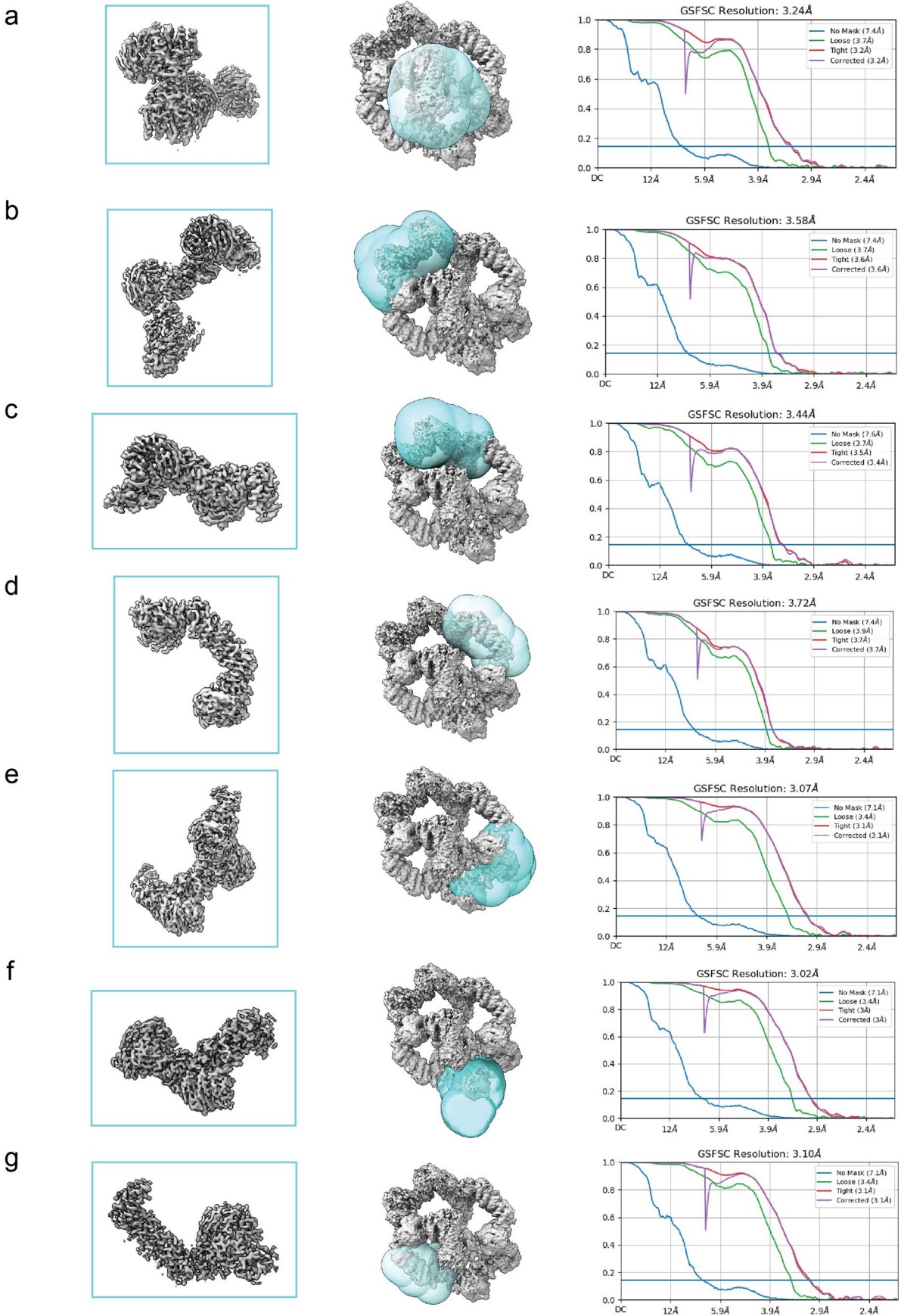

**Extended Data Fig. 3 | Local Refinement for GATOR2-CASTOR1. (a-g)** Local refinement for different sections of complex. Including mask (shown in cyan), FSC graph and resulting map.

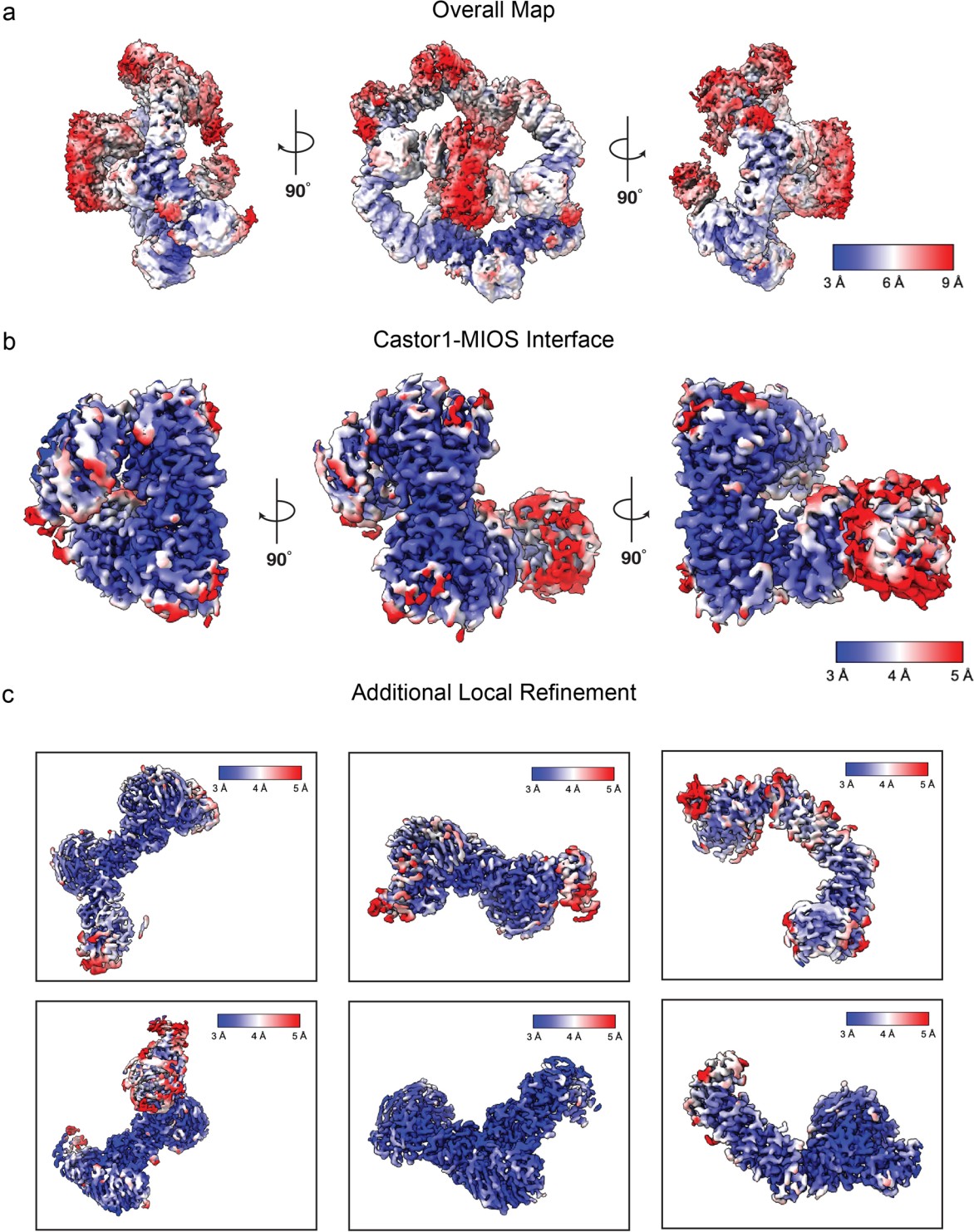

**Extended Data Fig. 4 | Local resolution estimation.** (**a**) Full complex map (**b**) CASTOR1-MIOS interface and (**c**) additional local refinement maps for the complex.

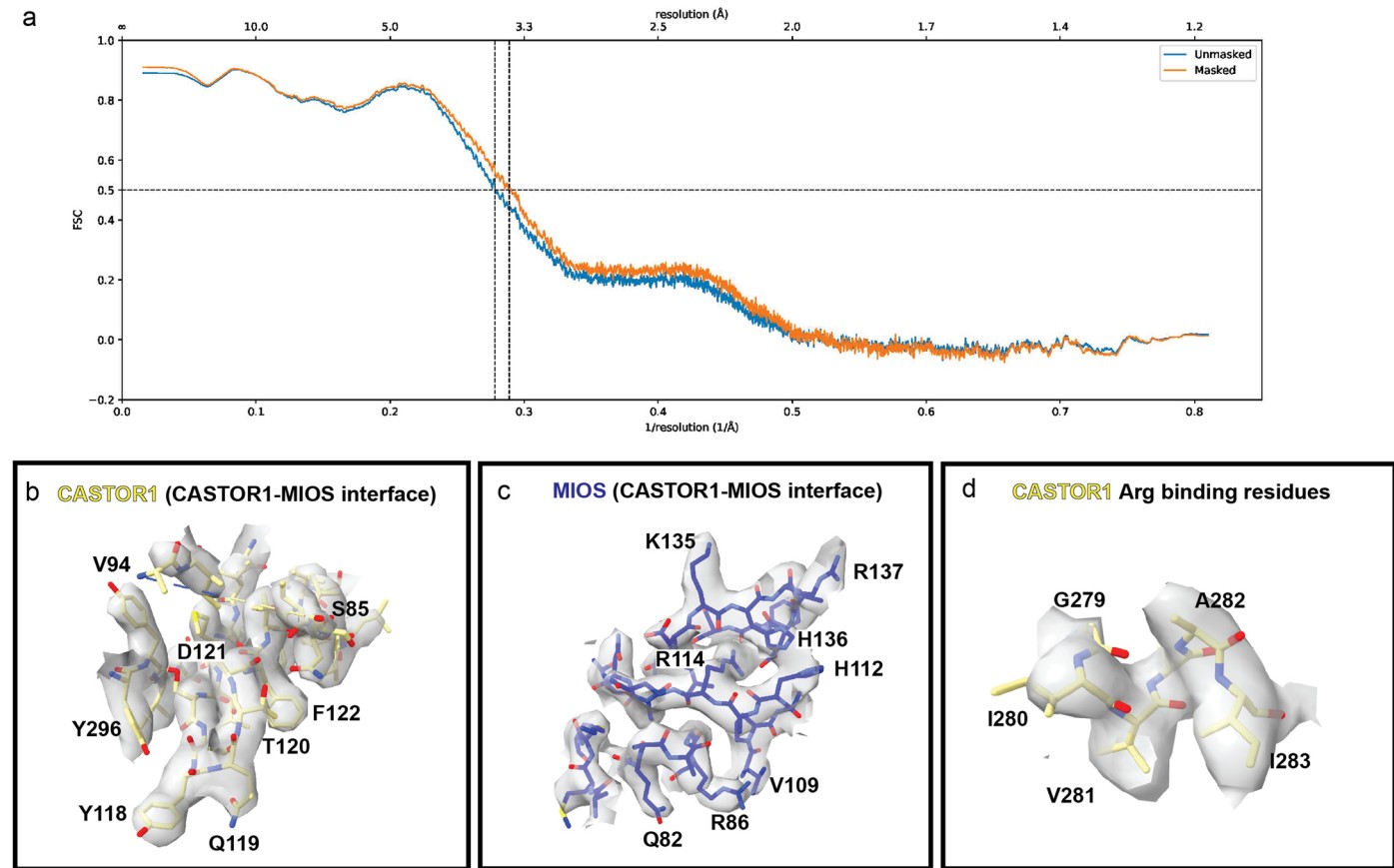

**Extended Data Fig. 5 | Map to model fit. (a)** Map-model FSC (**b**) CASTOR1 at CASTOR1-MIOS interface, (**c**) MIOS at CASTOR1-MIOS interface (**d**) CASTOR1 residues near arginine binding pocket.

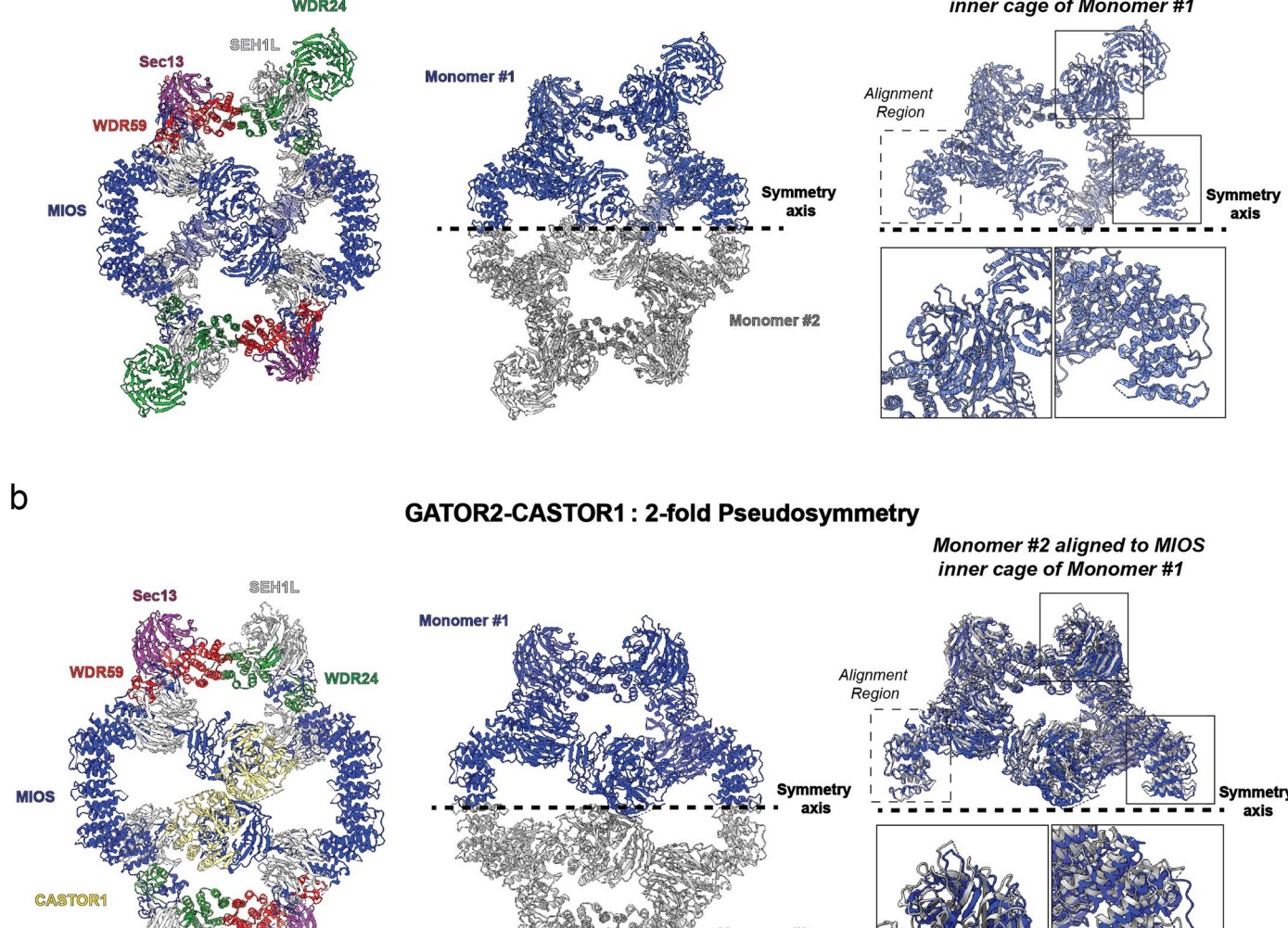

**Extended Data Fig. 6 | GATOR2 Cage Symmetry.** Comparison of cage symmetry for (**a**) GATOR2 unbound and (**b**) GATOR2-CASTOR1 complex. For each complex the individual monomers are reflected over the symmetry axis. Regions distal to the alignments region are enlarged for visualization.

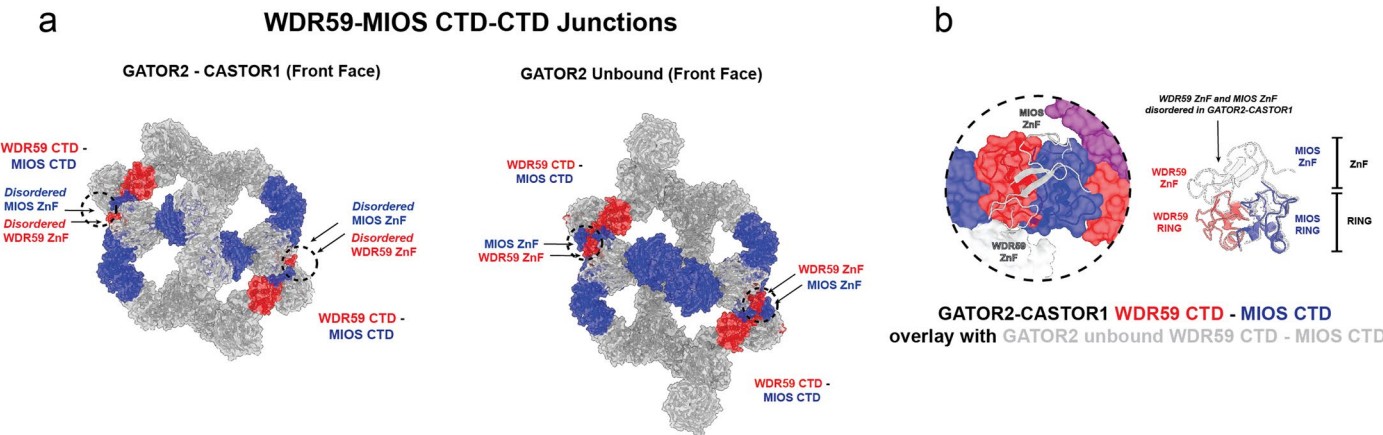

**Extended Data Fig. 7 | GATOR2 WDR59-MIOS CTD-CTD Junctions.** (**a**) Comparison of the WDR59-MIOS junctions (black dash circle) on GATOR2-CASTOR1 complex and. GATOR2 unbound. (**b**) Close up view of the changes to the WDR59-MIOS CTD junctions. GATOR2 unbound (grey) overlayed with the GATOR2-CASTOR1 (blue and red).

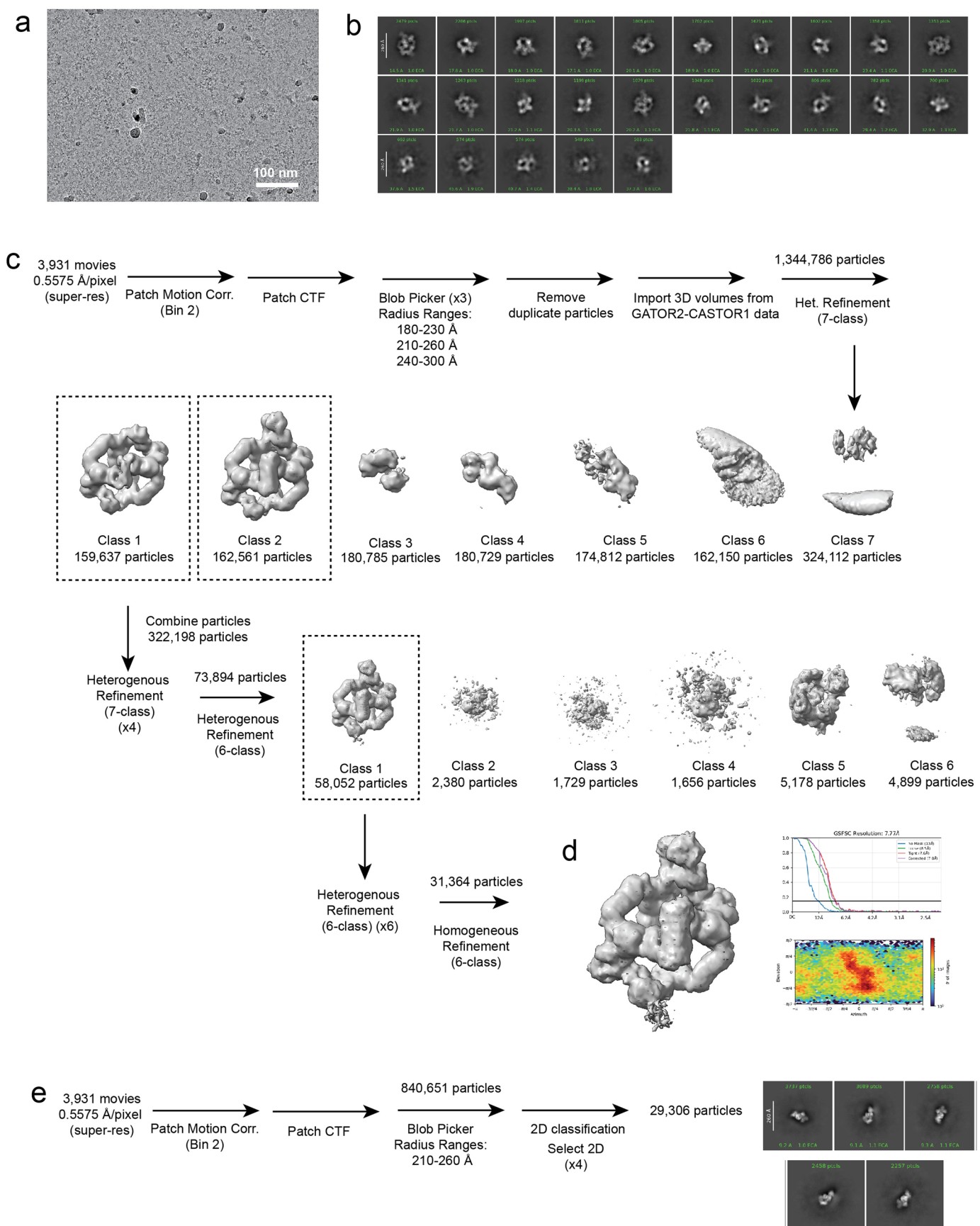

**Extended Data Fig. 8 | Data Processing Pipeline for GATOR2-CASTOR1-Sestrin2. (a)** Representative micrograph **(b)** Representative 2D classes **(c)** Data processing workflow **(d)** Overall map for GATOR2-CASTOR1, FSC graph and orientation plot. **(e)** Data processing for GATOR1 and representative 2D classes of isolated GATOR1 complex particles.

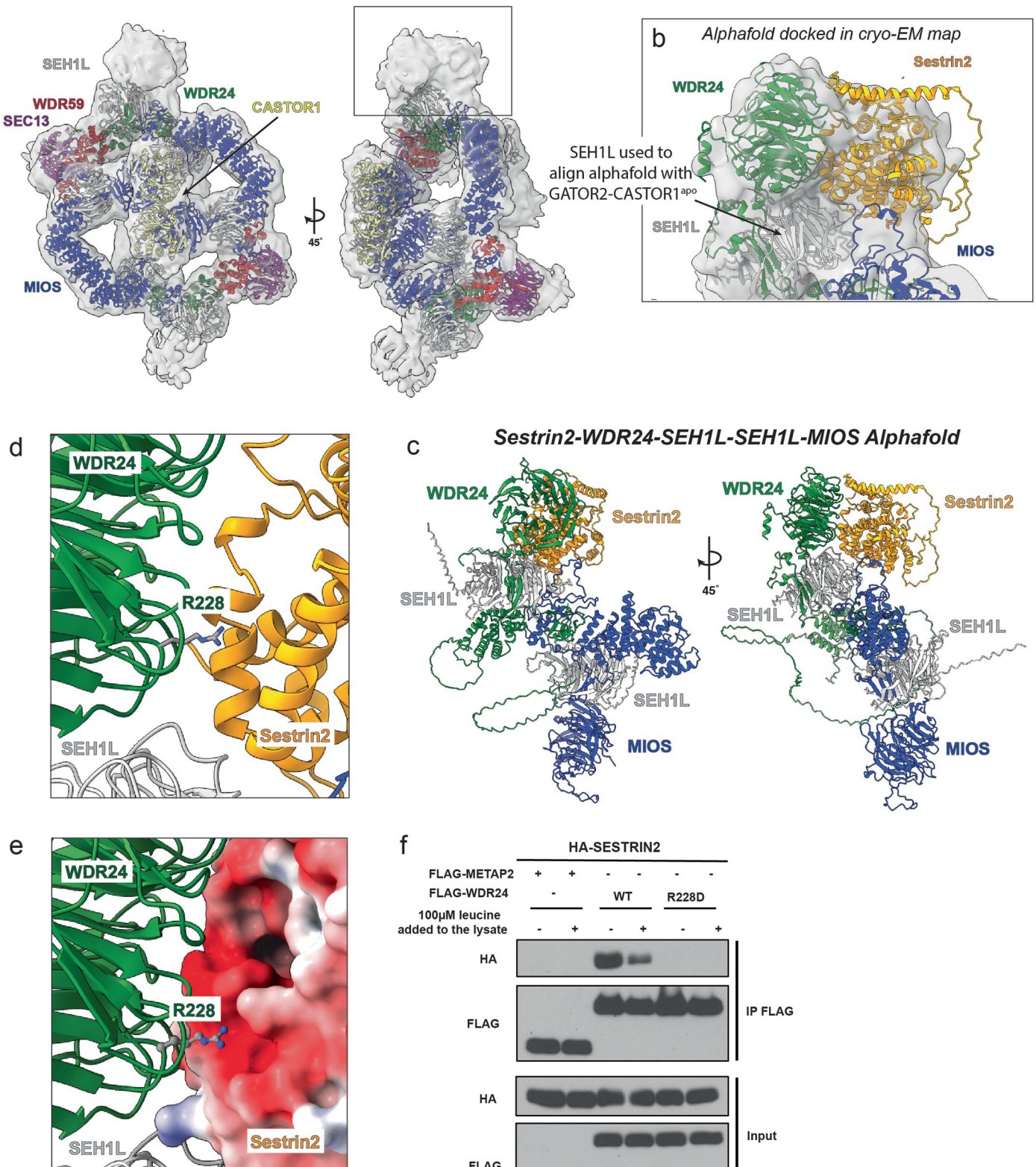

**Extended Data Fig. 9 | GATOR2-CASTOR1- Sestrin2 interaction. (a)** GATOR2-CASTOR1 structure docked into cryo-EM map of GATOR2-CASTOR1-Sestrin2. **(b)** Close up of GATOR2-CASTOR1-Sestrin2 cryo-EM density fitted with Sestrin2-WDR24-SEH1L-SEH1L-MIOS AlphaFold model **(c)** Full Sestrin2-WDR24-SEH1L-SEH1L-MIOS AlphaFold model (ipTM = 0.69). Close up of interface between WDR24 (green) and Sestrin2 (orange) in AlphaFold model in **(d)** ribbon view and **(e)** surface view colored by electrostatic potential. pLDDT for Arg 228 is 0.89.

**(f)** HEK-293T cells transiently expressing HA-tagged SESTRIN2 along with the indicated FLAG-tagged WDR24 constructs or FLAG-tagged METAP2 as a control were starved of leucine for 50 minutes. Where indicated, leucine was added to the lysates during immunoprecipitation. FLAG-immunoprecipitates were generated and analyzed by immunoblotting for the indicated proteins. Experiment performed twice with similar results.

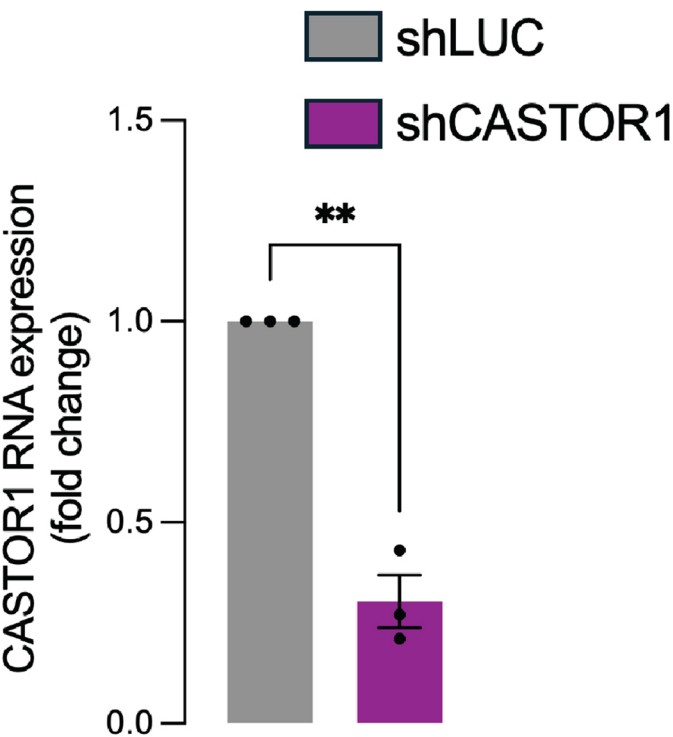

**Extended Data Fig. 10 | qPCR confirmation shCASTOR1.** qPCR against CASTOR1 performed in HEK293T transduced with a shRNA targeting Luciferase (shLUC) or a shRNA targeting CASTOR1. Data were normalized using ACTB and HPRT1 as housekeeping genes and are presented as the mean ± SEM of a biological triplicate. Unpaired two tailed t-test. ***p = 0.0004.

# Reporting Summary

## Statistics

For all statistical analyses, confirm that the following items are present in the figure legend, table legend, main text, or Methods section.

| n/a | Confirmed | |
|---|---|---|
| ☐ | ☒ | The exact sample size (*n*) for each experimental group/condition, given as a discrete number and unit of measurement |
| ☒ | ☐ | A statement on whether measurements were taken from distinct samples or whether the same sample was measured repeatedly |
| ☒ | ☐ | The statistical test(s) used AND whether they are one- or two-sided<br>*Only common tests should be described solely by name; describe more complex techniques in the Methods section.* |
| ☒ | ☐ | A description of all covariates tested |
| ☒ | ☐ | A description of any assumptions or corrections, such as tests of normality and adjustment for multiple comparisons |
| ☐ | ☒ | A full description of the statistical parameters including central tendency (e.g. means) or other basic estimates (e.g. regression coefficient) AND variation (e.g. standard deviation) or associated estimates of uncertainty (e.g. confidence intervals) |
| ☒ | ☐ | For null hypothesis testing, the test statistic (e.g. *F*, *t*, *r*) with confidence intervals, effect sizes, degrees of freedom and *P* value noted<br>*Give P values as exact values whenever suitable.* |
| ☒ | ☐ | For Bayesian analysis, information on the choice of priors and Markov chain Monte Carlo settings |
| ☒ | ☐ | For hierarchical and complex designs, identification of the appropriate level for tests and full reporting of outcomes |
| ☒ | ☐ | Estimates of effect sizes (e.g. Cohen's *d*, Pearson's *r*), indicating how they were calculated |

*Our web collection on statistics for biologists contains articles on many of the points above.*

## Software and code

Policy information about availability of computer code

| Data collection | SerialEM, CryoSPARC v4.3.1 and v4.4.1 |
|---|---|
| Data analysis | Chimera X, COOT 0.9.8. Statistical analysis was performed using GraphPad Prism Version 10.2.0 (335). |

For manuscripts utilizing custom algorithms or software that are central to the research but not yet described in published literature, software must be made available to editors and reviewers. We strongly encourage code deposition in a community repository (e.g. GitHub). See the Nature Portfolio guidelines for submitting code & software for further information.

## Data

Policy information about availability of data

All manuscripts must include a data availability statement. This statement should provide the following information, where applicable:

- Accession codes, unique identifiers, or web links for publicly available datasets
- A description of any restrictions on data availability
- For clinical datasets or third party data, please ensure that the statement adheres to our policy

The coordinates and density map have been deposited in the RCSB and EMDB with accession codes 9OTI and EMD-70833. Focused maps have been deposited in the EMDB with accession codes EMD-71136, EMD-71137, EMD-71138, EMD-71139, EMD-71140, EMD-71141, and EMD-71143.

# Field-specific reporting

Please select the one below that is the best fit for your research. If you are not sure, read the appropriate sections before making your selection.

☒ Life sciences ☐ Behavioural & social sciences ☐ Ecological, evolutionary & environmental sciences

For a reference copy of the document with all sections, see nature.com/documents/nr-reporting-summary-flat.pdf

# Life sciences study design

All studies must disclose on these points even when the disclosure is negative.

| | |
|---|---|
| Sample size | Sample sizes were not predetermined using statistical methods. The use of three replicates is customary in our field. |
| Data exclusions | No data were excluded from analysis |
| Replication | All experiments were successfully replicated three times, except for the extended figure 9f, which was replicated two times. |
| Randomization | Randomization is not considered customary or necessary in this field. |
| Blinding | Blinding  is not considered customary or necessary in this field. |

# Reporting for specific materials, systems and methods

We require information from authors about some types of materials, experimental systems and methods used in many studies. Here, indicate whether each material, system or method listed is relevant to your study. If you are not sure if a list item applies to your research, read the appropriate section before selecting a response.

## Materials & experimental systems

| n/a | Involved in the study |
|---|---|
| ☐ | ☒ Antibodies |
| ☐ | ☒ Eukaryotic cell lines |
| ☒ | ☐ Palaeontology and archaeology |
| ☒ | ☐ Animals and other organisms |
| ☒ | ☐ Human research participants |
| ☒ | ☐ Clinical data |
| ☒ | ☐ Dual use research of concern |

## Methods

| n/a | Involved in the study |
|---|---|
| ☒ | ☐ ChIP-seq |
| ☒ | ☐ Flow cytometry |
| ☒ | ☐ MRI-based neuroimaging |

## Antibodies

| | |
|---|---|
| Antibodies used | MIOS (Cell Signaling Technology, Cat#13557S, Clone#D12C6, Lot#1), WDR59 (Cell Signaling Technology, Cat#53385S, Clone#D4Z7A, Lot#1), FLAG (Cell Signaling Technology, Cat#14793S, Clone#D6W5B, Lot#7), HA (Cell Signaling Technology, Cat#3724S, Clone#C29F4, Lot#11), S6K1 (Cell Signaling Technology, Cat#2708S, Clone#49D7, Lot#8), phospho-T389-S6K1 (Cell Signaling Technology, Cat#9234S, Clone#108D2, Lot#16). Antibodies were used at the following dilutions: MIOS 1:1000 (Cell Signaling Technology, Cat#13557S, Clone#D12C6, Lot#1), WDR59 1:1000 (Cell Signaling Technology, Cat#53385S, Clone#D4Z7A, Lot#1), FLAG 1:1000 (Cell Signaling Technology, Cat#14793S, Clone#D6W5B, Lot#7), HA 1:1000 (Cell Signaling Technology, Cat#3724S, Clone#C29F4, Lot#11), S6K1 1:1000 (Cell Signaling Technology, Cat#2708S, Clone#49D7, Lot#8), phospho-T389-S6K1 1:1000 (Cell Signaling Technology, Cat#9234S, Clone#108D2, Lot#16). |
| Validation | MIOS (Cell Signaling Technology, Cat#13557S, Clone#D12C6, Lot#1), monoclonal antibody produced by immunizing animals with a synthetic peptide corresponding to residues surrounding Leu730 of human Mios protein. Recognizes endogenous levels of total Mios protein from Human, Mouse and Rat by Western blot (https://www.cellsignal.com/products/primary-antibodies/mios-d12c6-rabbit-mab/13557?srsltid=AfmBOopX3nLp8fdT5IsdpECXzToUYElh7UA95rwj0wb2zOec0FZb8gT8). |
| | WDR59 (Cell Signaling Technology, Cat#53385S, Clone#D4Z7A, Lot#1), monoclonal antibody produced by immunizing animals with a synthetic peptide corresponding to residues surrounding His356 of human WDR59 protein. Recognizes endogenous levels of total WDR59 protein from Human and Monkey by Western blot (https://www.cellsignal.com/products/primary-antibodies/wdr59-d4z7a-rabbit-mab/53385?srsltid=AfmBOoqJ0MRPgVUUZ8vXJJyriFfm-EUD_Fn11MA0R5k0Pu_c-EZOcN8F). |
| | FLAG (Cell Signaling Technology, Cat#14793S, Clone#D6W5B, Lot#7), monoclonal antibody produced by immunizing animals with a synthetic DYKDDDDK peptide. Detects exogenously expressed DYKDDDDK proteins in cells. By Western blot, the antibody recognizes the DYKDDDDK peptide, which is the same epitope recognized by Sigma-Aldrich Anti-FLAG M2 antibody, fused to either the amino-terminus or carboxy-terminus of the target protein (https://www.cellsignal.com/products/primary-antibodies/dykddddk-tag-d6w5b-rabbit-mab-binds-to-same-epitope-as-sigma-aldrich-anti-flag-m2-antibody/14793?srsltid=AfmBOor5TWqJlYICgfOXmddb9d7FC8x5rEF_EGz8sAKmj7yZlczklp-c). |

HA (Cell Signaling Technology, Cat#3724S, Clone#C29F4, Lot#11), monoclonal antibody produced by immunizing animals with a synthetic peptide containing the influenza hemagglutinin epitope (YPYDVPDYA). By western blot, detects exogenously expressed proteins containing the HA epitope tag. The antibody may cross-react with a protein of unknown origin ~100kDa (https://www.cellsignal.com/products/primary-antibodies/ha-tag-c29f4-rabbit-mab/3724?srsltid=AfmBOooWThRdxyoWIsodZBfJjx5jJ_Me--yDubOnEwlcJ8pLYc8F7kum).

S6K1 (Cell Signaling Technology, Cat#2708S, Clone#49D7, Lot#8), monoclonal antibody produced by immunizing animals with a synthetic peptide corresponding to residues surrounding the amino-terminus of human p70 S6 kinase. Detects endogenous levels of total p70 S6 kinase protein from Human by Western blot. The antibody also recognizes p85 S6 kinase (https://www.cellsignal.com/products/primary-antibodies/p70-s6-kinase-49d7-rabbit-mab/2708?srsltid=AfmBOoqKAFLvrXPGZDqJ0MH1Abfbij7FXPpaD09AdQP2YqXg-grA4fhp)

phospho-T389-S6K1 (Cell Signaling Technology, Cat#9234S, Clone#108D2, Lot#16), monoclonal antibody produced by immunizing animals with a synthetic phosphopeptide corresponding to residues surrounding Thr389 of human p70 S6 kinase. Detects endogenous levels of p70 S6 kinase only when phosphorylated at Thr389 from Human, Mouse, Rat and Mondey by Western blot. This antibody also detects p85 S6 kinase when phosphorylated at the analogous site (Thr412) and possibly S6KII phosphorylated at Thr388. This antibody may detect a non-specific band that runs around 62 kDa in some samples. The band is not phosphatase sensitive (https://www.cellsignal.com/products/primary-antibodies/phospho-p70-s6-kinase-thr389-108d2-rabbit-mab/9234?srsltid=AfmBOorVPkWzz4ryrCic7_SExN4cJbn2_8QaY6ydTEz5oo_b-omfCT2G)

MIOS (Cell Signaling Technology, Cat#13557S, Clone#D12C6, Lot#1), monoclonal antibody produced by immunizing animals with a synthetic peptide corresponding to residues surrounding Leu730 of human Mios protein. Recognizes endogenous levels of total Mios protein from Human, Mouse and Rat by Western blot (https://www.cellsignal.com/products/primary-antibodies/mios-d12c6-rabbit-mab/13557?srsltid=AfmBOopX3nLp8fdT5IsdpECXzToUYElh7UA95rwj0wb2zOec0FZb8gT8).

WDR59 (Cell Signaling Technology, Cat#53385S, Clone#D4Z7A, Lot#1), monoclonal antibody produced by immunizing animals with a synthetic peptide corresponding to residues surrounding His356 of human WDR59 protein. Recognizes endogenous levels of total WDR59 protein from Human and Monkey by Western blot (https://www.cellsignal.com/products/primary-antibodies/wdr59-d4z7a-rabbit-mab/53385?srsltid=AfmBOoqJ0MRPgVUUZ8vXJJyriFfm-EUD_Fn11MA0R5k0Pu_c-EZOcN8F).

FLAG (Cell Signaling Technology, Cat#14793S, Clone#D6W5B, Lot#7), monoclonal antibody produced by immunizing animals with a synthetic DYKDDDDK peptide. Detects exogenously expressed DYKDDDDK proteins in cells. By Western blot, the antibody recognizes the DYKDDDDK peptide, which is the same epitope recognized by Sigma-Aldrich Anti-FLAG M2 antibody, fused to either the amino-terminus or carboxy-terminus of the target protein (https://www.cellsignal.com/products/primary-antibodies/dykddddk-tag-d6w5b-rabbit-mab-binds-to-same-epitope-as-sigma-aldrich-anti-flag-m2-antibody/14793?srsltid=AfmBOor5TWqJlYICgfOXmddb9d7FC8x5rEF_EGz8sAKmj7yZlczklp-c).

HA (Cell Signaling Technology, Cat#3724S, Clone#C29F4, Lot#11), monoclonal antibody produced by immunizing animals with a synthetic peptide containing the influenza hemagglutinin epitope (YPYDVPDYA). By western blot, detects exogenously expressed proteins containing the HA epitope tag. The antibody may cross-react with a protein of unknown origin ~100kDa (https://www.cellsignal.com/products/primary-antibodies/ha-tag-c29f4-rabbit-mab/3724?srsltid=AfmBOooWThRdxyoWIsodZBfJjx5jJ_Me--yDubOnEwlcJ8pLYc8F7kum).

S6K1 (Cell Signaling Technology, Cat#2708S, Clone#49D7, Lot#8), monoclonal antibody produced by immunizing animals with a synthetic peptide corresponding to residues surrounding the amino-terminus of human p70 S6 kinase. Detects endogenous levels of total p70 S6 kinase protein from Human by Western blot. The antibody also recognizes p85 S6 kinase (https://www.cellsignal.com/products/primary-antibodies/p70-s6-kinase-49d7-rabbit-mab/2708?srsltid=AfmBOoqKAFLvrXPGZDqJ0MH1Abfbij7FXPpaD09AdQP2YqXg-grA4fhp)

phospho-T389-S6K1 (Cell Signaling Technology, Cat#9234S, Clone#108D2, Lot#16), monoclonal antibody produced by immunizing animals with a synthetic phosphopeptide corresponding to residues surrounding Thr389 of human p70 S6 kinase. Detects endogenous levels of p70 S6 kinase only when phosphorylated at Thr389 from Human, Mouse, Rat and Mondey by Western blot. This antibody also detects p85 S6 kinase when phosphorylated at the analogous site (Thr412) and possibly S6KII phosphorylated at Thr388. This antibody may detect a non-specific band that runs around 62 kDa in some samples. The band is not phosphatase sensitive (https://www.cellsignal.com/products/primary-antibodies/phospho-p70-s6-kinase-thr389-108d2-rabbit-mab/9234?srsltid=AfmBOorVPkWzz4ryrCic7_SExN4cJbn2_8QaY6ydTEz5oo_b-omfCT2G)

# Eukaryotic cell lines

Policy information about cell lines

| | |
|---|---|
| Cell line source(s) | Human Embryonic Kidney cells (HEK-293T) were obtained from ATCC (https://www.atcc.org/). |
| Authentication | HEK-293T was verified by ATCC (https://www.atcc.org/products/crl-3216), as well as by morphological analysis. |
| Mycoplasma contamination | All cel lines used in this study were tested negative for mycoplasma contamination usng MycoAlert Mycoplasma Detection kit (Lonza, LT-07-318). |
| Commonly misidentified lines (See ICLAC register) | No ICLAC cell lines were used in this study |

