## [Peer Review File · Nature Structural & Molecular Biology]

Structural basis for mTORC1 regulation by the CASTOR1-GATOR2 complex

Corresponding Author: Professor James Hurley

Version 0:

Decision Letter:

Our ref: NSMB-A50968-T

29th May 2025

Dear Jim,

I am writing on Kat's behalf as she is out of the office.

Thank you for submitting your revised manuscript "Structural basis for mTORC1 regulation by the CASTOR1-GATOR2 complex" (NSMB-A50968-T). It has now been seen by the original referee Rev#2 and their comments are below. We asked Rev#2 to help us assess changes made in response to Rev#1's points as well. The reviewer finds that the paper has improved in revision, and therefore we'll be happy in principle to publish it in Nature Structural & Molecular Biology, pending minor revisions to satisfy the referee's final requests and to comply with our editorial and formatting guidelines.

We are now performing detailed checks on your paper and will send you a checklist detailing our editorial and formatting requirements in about 1-2 weeks. Please do not upload the final materials and make any revisions until you receive this additional information from us.

Thank you again for your interest in Nature Structural & Molecular Biology. Please do not hesitate to contact me if you have any questions.

Sincerely,

Melina Casadio, PhD
Locum Chief Editor, Nature Structural & Molecular Biology
ORCID ID: <https://orcid.org/0000-0003-2389-2243>

Reviewer #2 (Remarks to the Author):

The authors have addressed my only (minor) comment. The revised manuscript merits publication in N.S.M.B.
